# Beiging of perivascular adipose tissue regulates its inflammation and vascular remodeling

Yusuke Adachi [1,2,7], Kazutaka Ueda [1,7] ✉, Seitaro Nomura[1,3], Kaoru Ito [4], Manami Katoh [1,3], Mikako Katagiri [1], Shintaro Yamada[1,3], Masaki Hashimoto[1], Bowen Zhai[1], Genri Numata [1], Akira Otani[1], Munetoshi Hinata[5], Yuta Hiraike [6], Hironori Waki[6], Norifumi Takeda[1], Hiroyuki Morita[1], Tetsuo Ushiku[5], Toshimasa Yamauchi[6], Eiki Takimoto[1] & Issei Komuro [1] ✉

Although inflammation plays critical roles in the development of atherosclerosis, its regulatory mechanisms remain incompletely understood. Perivascular adipose tissue (PVAT) has been reported to undergo inflammatory changes in response to vascular injury. Here, we show that vascular injury induces the beiging (brown adipose tissue-like phenotype change) of PVAT, which fine-tunes inflammatory response and thus vascular remodeling as a protective mechanism. In a mouse model of endovascular injury, macrophages accumulate in PVAT, causing beiging phenotype change. Inhibition of PVAT beiging by genetically silencing PRDM16, a key regulator to beiging, exacerbates inflammation and vascular remodeling following injury. Conversely, activation of PVAT beiging attenuates inflammation and pathological vascular remodeling. Single-cell RNA sequencing reveals that beige adipocytes abundantly express neuregulin 4 (*Nrg4*) which critically regulate alternative macrophage activation. Importantly, significant beiging is observed in the diseased aortic PVAT in patients with acute aortic dissection. Taken together, vascular injury induces the beiging of adjacent PVAT with macrophage accumulation, where NRG4 secreted from the beige PVAT facilitates alternative activation of macrophages, leading to the resolution of vascular inflammation. Our study demonstrates the pivotal roles of PVAT in vascular inflammation and remodeling and will open a new avenue for treating atherosclerosis.

Inflammation is deeply involved in the development of cardiovascular diseases, particularly in atherosclerosis as responses to vascular injury[1-3]. Temporal inflammation and its resolution are required for

wound healing and repair after vascular injury, and prolonged excessive inflammation causes pathological vascular remodeling that manifests as neointimal hyperplasia[4,5]. It is widely accepted that the

---

[1]Department of Cardiovascular Medicine, Graduate School of Medicine, The University of Tokyo, Tokyo, Japan. [2]Department of Advanced Clinical Science and Therapeutics, The University of Tokyo, Tokyo, Japan. [3]Genome Science Division, Research Center for Advance Science and Technology, The University of Tokyo, Tokyo, Japan. [4]Laboratory for Cardiovascular Genomics and Informatics, RIKEN Center for Integrative Medical Sciences, Yokohama, Kanagawa, Japan. [5]Department of Pathology, Graduate School of Medicine, The University of Tokyo, Tokyo, Japan. [6]Department of Diabetes and Metabolic Diseases, Graduate School of Medicine, The University of Tokyo, Tokyo, Japan. [7]These authors contributed equally: Yusuke Adachi, Kazutaka Ueda. ✉e-mail: uedak-tky@umin.ac.jp; komuro-tky@umin.ac.jp

pathogenesis of vascular remodeling starts from the impaired endo-thelial cell function, leading to recruitment of circulating inflammatory cells to the intima–medial layer[3,6,7]. The effects of potent vascular protective agents such as estrogen, statins and microRNAs have been mostly attributed to their beneficial functions on endothelial cells[8–12]. There is also growing evidence suggesting that perivascular adipose tissue (PVAT), which accounts for the majority of the outer matrix surrounding systemic blood vessels, contributes to the pathophysio-logical vascular response in an 'outside-in' manner[13]. Adipose tissues are classified into two distinct phenotypes; white adipose tissue (WAT), which stores energy as triglycerides, and brown adipose tissue (BAT), which dissipates energy as heat[14]. A brown adipocyte-like phe-notype has been reported to emerge within WAT in response to var-ious stimuli such as cold exposure and β3-adrenergic receptor (β3AR) agonist[14,15]. This unique phenomenon, called 'browning' or 'beiging', is controlled by transcriptional regulators, such as *PRDM16*, *NFIA* and *EBF2*[16–22], and observed in humans as well as in rodents, contributing to energy homeostasis[14,15]. Recent basic and clinical studies reported that human PVAT surrounding coronary arteries show phenotypic changes in response to vascular inflammation, which can be detected by newly innovated computerized tomography-based methods[23,24]. Although these findings suggest an association between PVAT phenotypic changes and vascular diseases, the causality and mechanisms involved remain unclear.

In this work, we examined the roles of PVAT in vascular inflam-mation and subsequent pathological remodeling. We show that vas-cular injury induces the beiging of PVAT, which fine-tunes inflammatory response and thus vascular remodeling as a protective mechanism.

## Results

### Endovascular injury induces PVAT beiging
We generated endovascular injury by inserting a wire into the femoral arteries (FAs) and examined inflammatory response in PVAT. Given the considerable effects of estrogen on the vasculature[25,26], male or ovar-iectomized female mice were used in the study. Genes of various types of immune cells were upregulated in PVAT at 24 h after vascular injury, including the monocyte-macrophage marker genes, *Cd11b*, *Cd11c*, *Mrc1*, and *Adgre1* (F4/80) (Fig. 1a). Immunohistochemical staining revealed that F4/80+ macrophages accumulated predominantly in the outer tissues surrounding the vasculature, which were mainly com-prised of PVAT, rather than in the vascular tissues, at the early phase (3 days) after injury (Fig. 1b, c). These F4/80+ macrophages then spread into the vascular tissues at the late phase (14 days) after injury (Fig. 1b, c). Concurrently, the expressions of inflammatory cytokine genes such as *Tnf* (TNF-α), *Serpine1* (PAI-1), *Il1a* and *Il1b*, were significantly upre-gulated in the surrounding tissues after vascular injury (Fig. 1d).

Transcriptome analysis of PVAT surrounding injured or sham-operated FAs showed that in addition to the increase in expression levels of several genes encoding immune cell markers and inflamma-tory cytokines, vascular injury induced a significant increase in expression levels of BAT marker genes including *Ucp1*, *Cidea*, *Cox8b*, *Ppargc1a*, *Elovl3* and *Dio2* in the PVAT (Supplementary Fig. 1a), which was confirmed by real-time quantitative reverse transcription poly-merase chain reaction (qRT-PCR) analysis (Fig. 1e). Upregulation of UCP1 in PVAT after vascular injury was confirmed by in situ hybridi-zation analysis (Fig. 1f) and immunohistochemical analysis (Fig. 1g, h). Morphologically, adipocytes in PVAT after vascular injury possessed the beige/BAT characteristics including smaller cell size and multi-locular lipid droplets compared with those after sham operation (Fig. 1g, i). Western blot analysis also showed the increase in UCP1 exclusively in the outer tissues denuded from vessels rather than the remained arteries (Fig. 1j). Then, we examined the role of infiltrated macrophages in the upregulation of UCP1. Reduction of macrophages by the administration of clodronate liposome significantly attenuated upregulation of UCP1 after vascular injury (Fig. 1k, l). The effects of

ovariectomy on the expression levels of BAT markers, such as *Ucp1* and *Elovl3*, were marginal, while the upregulation of these markers after vascular injury was more prominent in ovariectomized mice than in controls (Supplementary Fig. 1b). A significant upregulation of UCP1 and the morphological shrinkage of adipocyte cell size in PVAT after vascular injury were observed in intact gonadal male mice (Supple-mentary Fig. 1c, d). Taken together, these results suggest that in male and ovariectomized female mice, vascular injury elicits the infiltration of macrophages in PVAT, which induces the change of PVAT into BAT-like phenotypes, resembling the "beiging"-phenomenon that is observed in subcutaneous and visceral WAT following exposure to cold or β3AR agonists.

### PVAT beiging alters vascular inflammatory response
We next investigated the pathophysiological roles of the PVAT beiging in the development of vascular remodeling after injury using mice with an adipocyte-specific deletion of *Prdm16* gene, an essential regulator of beiging[16–18], using *Adipoq* (Adiponectin)–Cre+/−;*Prdm16*flox/flox mice, referred to as AdipoCre+;*Prdm16*. *Prdm16* mRNA expression was >90% lower in adipose tissues such as thoracic aorta PVAT and intra-scapular BAT compared with AdipoCre−;*Prdm16* control littermates, despite comparable expression in other tissues such as lung and intestine (Supplementary Fig. 2a). Vascular injury induced more prominent remodeling accompanied by marked inflammation and suppressed beiging in the PVAT in both males and ovariectomized females of AdipoCre+;*Prdm16* mice compared with AdipoCre−;*Prdm16* mice at 14 days after injury (Fig. 2a, b and Supplementary Fig. 2b, c). To rule out the effects of *Prdm16* deletion in adipose tissues of the whole body, we locally repressed *Prdm16* by applying pluronic gel containing small interfering RNA (siRNA) against *Prdm16* around FAs[27,28]. A significant reduction of *Prdm16* mRNA expression was confirmed in the outer tissues surrounding FA of wild type mice after siRNA treatment, whereas no difference was observed in contralateral untreated FA (Supplementary Fig. 2d). The *Prdm16* siRNA treatment abolished PVAT beiging after vascular injury (Supplementary Fig. 2e) and exacerbated pathological intimal thickening 14 days after vascular injury (Fig. 2c).

Macrophages change their phenotypical and functional char-acteristics, termed macrophage polarization, in response to pro- or anti-inflammatory stimuli[29]. Although the accumulation of F4/80+ cells was comparable between locally beiging-suppressed FA and control FA (Fig. 2c), the ratio of inducible nitric oxide synthase (iNOS)+ activated inflammatory macrophages to CD206+ alternatively activated anti-inflammatory macrophages was significantly increased in the beiging-suppressed FA PVAT as compared with that of control FA PVAT 14 days after vascular injury (Fig. 2c). These significant differences were not observed during the early phase (3 days) after injury (Supplementary Fig. 3a, b), suggesting that inhibition of PVAT beiging exacerbates pathological remodeling after vascular injury by prolonged activation of pro-inflammatory macrophages.

The effects of systemic activation of WAT beiging by exposure to cold or β3AR agonists on vascular pathogenesis in vivo remain con-troversial and depend on the experimental models studied[30,31]. These conflicting results are considered to come from direct and/or indirect effects of systemic activation of beiging on vascular remodeling, such as brown/beige fat-mediated lipolysis, release of anti-inflammatory adipokines and glucose homeostasis[30–32]. Therefore, we locally acti-vated beiging of the FA PVAT using pluronic gel containing the specific β3AR agonist, CL316243 and evaluated the specific effects of PVAT beiging on the phenotypic transition of macrophage and vascular remodeling. Local β3AR activation significantly induced PVAT beiging in the outer surrounding tissue of FA, but not in the contralateral untreated FA, visceral WAT or BAT (Supplementary Fig. 3c), and sig-nificantly attenuated vascular intimal thickening 14 days after injury with a shift of macrophage polarization to anti-inflammatory pheno-type in FA PVAT (Fig. 2d and Supplementary Fig. 3d). These significant

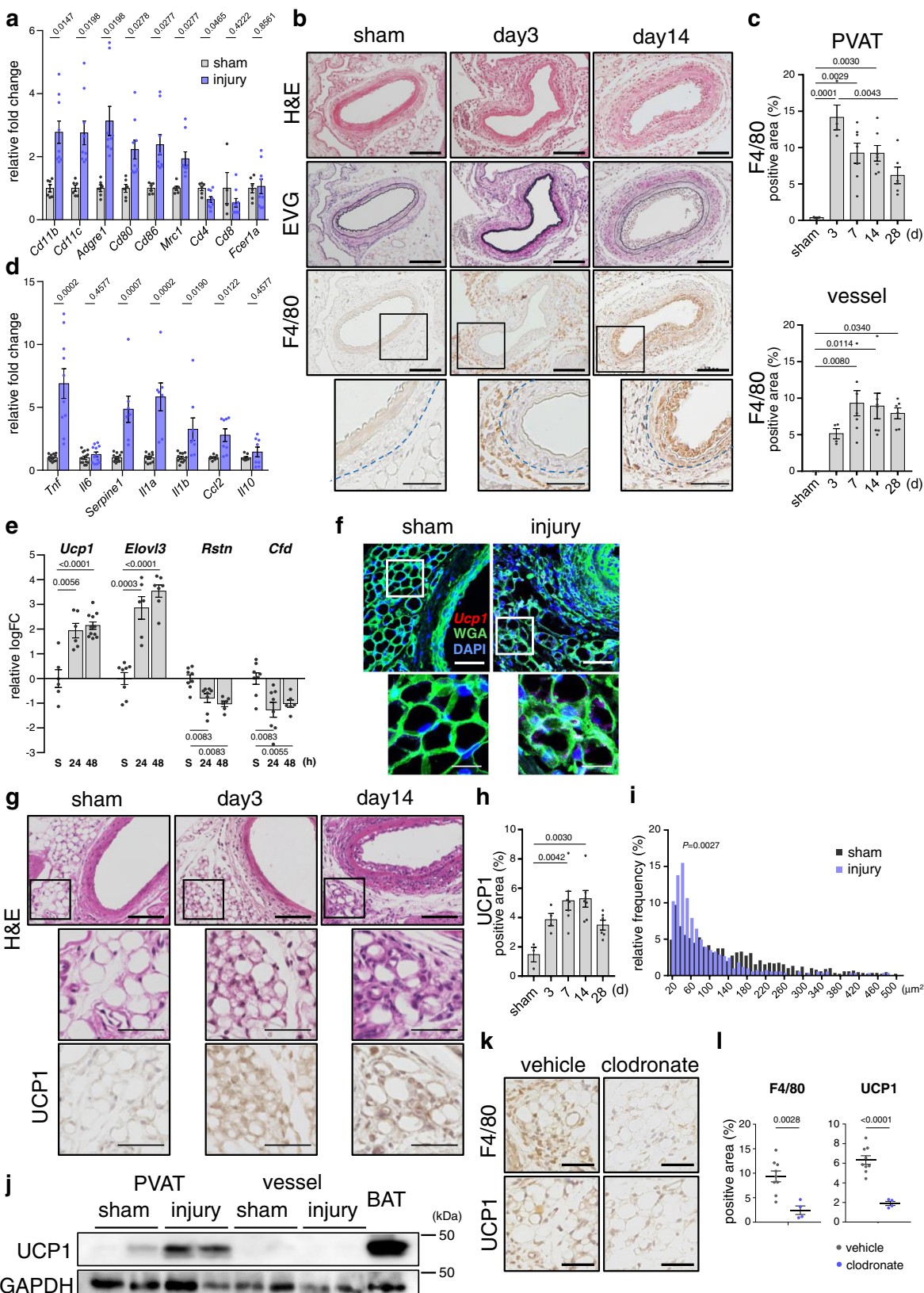

differences were not observed during the early phase after injury (Supplementary Fig. 3a, b). In contrast, the beiging in PVAT elicited by vascular injury was attenuated by a β3AR antagonist (SR59230A) with pluronic gel (Supplementary Fig. 3e). These data suggest that pharmacological beiging stimulation exerts additional protection against pathological remodeling after vascular injury by suppression of

prolonged activation of pro-inflammatory macrophages, and that the beiging in PVAT after the injury is mediated by the β3AR signaling.

Then, we examined the interactions between PVAT-derived beige adipocytes and macrophages. When RAW 264.7 macrophage cells or bone marrow-derived macrophages (BMDMs) were cultured in medium conditioned by beige adipocytes differentiated from the stromal

**Fig. 1 | Endovascular injury-induced macrophage accumulation and inflammation in PVAT followed by upregulation of BAT markers. a** Gene expressions of immune cell markers in PVAT 24 h after vascular injury (sham, $n = 4$ for *Cd8*, 6 for others; injury, $n = 9$, two-tailed $t$ tests with Holm-Sidak's correction for multiple comparisons). **b** Haematoxylin and eosin (H&E), Elastica van Gieson (EVG) and immunohistochemical staining for F4/80 in the early (day 3) and late (day 14) phases after injury. Blue dashed lines indicate the external elastic lamina. Scale bars represent 100 μm (thick bars) and 50 μm (thin bars). Images are representative of three independent experiments. **c** Time course of F4/80⁺ cell accumulation in arteries (vessel) and outer tissues mainly composed of PVAT ($n = 3, 4, 7, 7, 6$ (from left to right) at each group, respectively, one-way analysis of variance (ANOVA) and Tukey–Kramer post-hoc test). **d** Gene expressions of inflammatory cytokine markers in PVAT 48 h after vascular injury (sham, $n = 12, 14, 12, 12, 12, 8, 8$, respectively; injury, $n = 10, 12, 7, 7, 8, 8, 8$, respectively, two-tailed $t$ tests with Holm-Sidak's correction for multiple comparisons. **e** Gene expressions of BAT (*Ucp1* and *Elovl3*) and WAT [*Rstn* (Resistin) and *Cfd* (Adipsin)] markers in PVAT at 24 and 48 h after injury (sham, $n = 6, 8, 8, 8$, respectively; injury 24 h, $n = 6, 6, 8, 8$, respectively; injury 48 h, $n = 11, 7, 6, 6$, respectively, one-way ANOVA followed by Tukey–Kramer post-hoc test). **f** In situ hybridization showing *Ucp1* mRNA expression 14 days after vascular injury in wild type mice. Scale bars represent 50 μm (thick bars) and 20 μm (thin bars). Images are representative of three independent experiments. **g** H&E

and immunohistochemical staining for UCP1 in injured (3 and 14 days after injury) or sham-operated FAs and outer tissue. Scale bars represent 100 μm (thick bars) and 50 μm (thin bars). Images are representative of three independent experiments. **h** Time course of UCP1⁺ cells accumulation in PVAT ($n = 3, 4, 7, 7, 6$ at each group, respectively, one-way ANOVA followed by Tukey–Kramer post-hoc test). **i** Histograms of adipocyte area of sham-operated or injured PVAT (14 days after injury). Three images of each biological replicate were analyzed and combined to create the histogram. The size distribution between each group was compared using Kolmogorov-Smirnov test. Each bin was normalized to a percent of the total count for that individual tissue. Adipocytes of the bin size in the range of 20-500 μm² were included for the analysis. **j** Representative images of western blots for UCP1 in arteries (vessel) and outer tissues mainly composed of PVAT harvested 48 h after injury. Intra-scapular BAT was used as a positive control. GAPDH was used for internal control ($n = 4$ for each group, representative images are shown). **k** Immunohistochemical staining for F4/80 and UCP1 in outer tissue surrounding FAs 14 days after injury was performed in mice treated with either clodronate or vehicle. Scale bars represent 50 μm. Images are representative of three independent experiments. **l** F4/80⁺ and UCP1⁺ area in PVAT (vehicle, $n = 9$; clodronate, $n = 4$, unpaired two-tailed Student's $t$ test). Data represent mean ± SEM. Source data are provided as a Source Data file.

vascular fraction of PVAT, expression levels of *Mrc1* (CD206) were increased and those of inflammatory cytokines such as *Tnf*, *Serpine1*, and *Il1b* were decreased (Fig. 3a-d). These changes were abrogated by inhibition of PVAT beiging by siRNA knockdown of *Prdm16* (Fig. 3e, f and Supplementary Fig. 3f). We further examined the effects of PVAT beiging on cell growth of macrophages. Culture medium conditioned by PVAT-derived beige adipocytes significantly decreased the number of RAW 264.7 macrophages activated by interferon-gamma (IFNγ) and lipopolysaccharide (LPS), but not that of macrophages alternatively activated by interleukin (IL) 4 (Fig. 3g and Supplementary Fig. 3g). Culture medium conditioned by PVAT-derived adipocytes with knockdown of *Prdm16* did not decrease the number of classically activated macrophages induced by IFNγ and LPS (Fig. 3g). Taken together, these results suggest that factors secreted from PVAT-derived beige adipocytes selectively inhibit the growth of classically activated macrophages and shift macrophage phenotypes into an alternatively activated state, resulting in anti-inflammatory effects.

## NRG4 as an anti-inflammatory factor

Some studies have shown the contributions of UCP1 to atherogenesis[30,32–35]. Therefore, we first examined UCP1 as a possible regulator of vascular remodeling after injury. Unexpectedly, vascular wall thickening 14 days after injury was comparable between *Ucp1⁻/⁻* and littermate wild type (*Ucp1⁺/⁺*) mice (Fig. 3h), suggesting that UCP1 is not involved in the beneficial effects of PVAT beiging on vascular remodeling.

To identify the key factors that control endovascular injury-induced inflammation in PVAT, we performed transcriptional RNA profiling using single-cell RNA sequencing (scRNA-seq) and subsequent computational analysis using publicly available scRNA-seq datasets of murine inguinal WAT (iWAT) treated with CL316243 or control (GSE 133486)[36]. Cluster analysis of uniform manifold approximation and projection (UMAP) dimensionality reduction after integration with Seurat revealed that the cells of iWAT were classified into 15 clusters (C0–14) and that C7(Seurat) mostly consisted of CL316243-treated cells (Fig. 4a, b and Supplementary Fig. 4a). Beige/brown adipocyte-specific genes such as *Ucp1*, *Cidea* and *Ppargc1a*, were predominantly expressed in C7(Seurat) (Fig. 4c and Supplementary Figs. 4b, 5), indicating that this cluster represents beige adipocytes induced by β3AR activation. The emergence of the beige adipocyte cluster by β3AR activation with CL316243 was also confirmed using another integration method, Harmony (C6(Harmony), Supplementary Figs. 6a–f and 7). In addition, the cell type deconvolution analysis of the bulk RNA-seq dataset of iWAT (GSE129083)[37] showed that the beige

adipocyte population appeared in iWAT treated with CL316243 (Supplementary Table 2). Furthermore, the analysis of another dataset (GSE 133486)[36] demonstrated that β3AR activation with cold stimuli also induced the emergence of a beige adipocyte population (C9(cold)), Supplementary Figs. 8a–e and 9). Gene ontology analysis for the significantly regulated genes in C7(Seurat) identified *Nrg4* as a highly expressed secretory factor involved in the molecular function of receptor-binding (Fig. 4c, d). The enrichment of *Nrg4* was also detected in C6(Harmony) and C9(cold) (Supplementary Figs. 6d, e and 8c, d).

Consistent with the findings from the comprehensive data analysis, we observed that *Nrg4* was upregulated in PVAT after vascular injury (Fig. 4e) and in cultured PVAT-derived preadipocytes stimulated by beige adipocyte differentiation factors, accompanied by other beige genes in a time-dependent manner (Fig. 4f). The upregulation of *Nrg4* in PVAT surrounding injured FAs was also observed 14 days after injury by in situ hybridization analysis (Supplementary Fig. 10a), where *Nrg4* was expressed in beige adipocytes of PVAT expressing *Ucp1* (Supplementary Fig. 10b). *Nrg4* upregulation was significantly attenuated in Adipo^Cre+;*Prdm16* mice and PVAT-derived preadipocytes transduced with siRNA against *Prdm16* or *Nfia* (Fig. 4g, h), whereas it occurred independently of UCP1 (Supplementary Fig. 10c). NRG4 has recently emerged as a brown fat-enriched secreted factor that ameliorates diet-induced metabolic disorders, including insulin resistance and hepatic steatosis[38–40]. We thus examined whether NRG4 was involved in beige adipocytes-induced resolution of macrophage inflammation. Culture media conditioned by PVAT-derived beige adipocytes induced phenotypic changes from classical to alternative activation and downregulation of inflammatory genes in RAW 264.7 macrophages or BMDMs, but these changes were not observed by the culture media conditioned by PVAT-derived beige adipocytes with knockdown of *Nrg4* by siRNA (Supplementary Fig. 10d and Fig. 4i–l). Consistently, recombinant murine NRG4 attenuated the *Cd86/Mrc1* ratio and inflammatory cytokine levels in macrophages classically activated (Supplementary Fig. 10e, f). The knockdown of Erb-B2 receptor tyrosine kinase 4 (ErbB4), a receptor of NRG4[41], in macrophages abolished the alternative activation and anti-inflammatory effects of culture media conditioned with PVAT-derived beige adipocytes (Supplementary Fig. 10g–i). Furthermore, culture media conditioned by PVAT-derived beige adipocytes with *Nrg4* knockdown no longer inhibited the increase in the number of RAW 264.7 cells activated by IFNγ and LPS (Fig. 4m). In mice, *Nrg4* siRNA treatment with pluronic gel was found to exacerbate pathological intimal thickening 14 days after vascular injury (Fig. 4n). Taken together, these results suggest that NRG4 secreted from PVAT-derived beige adipocytes

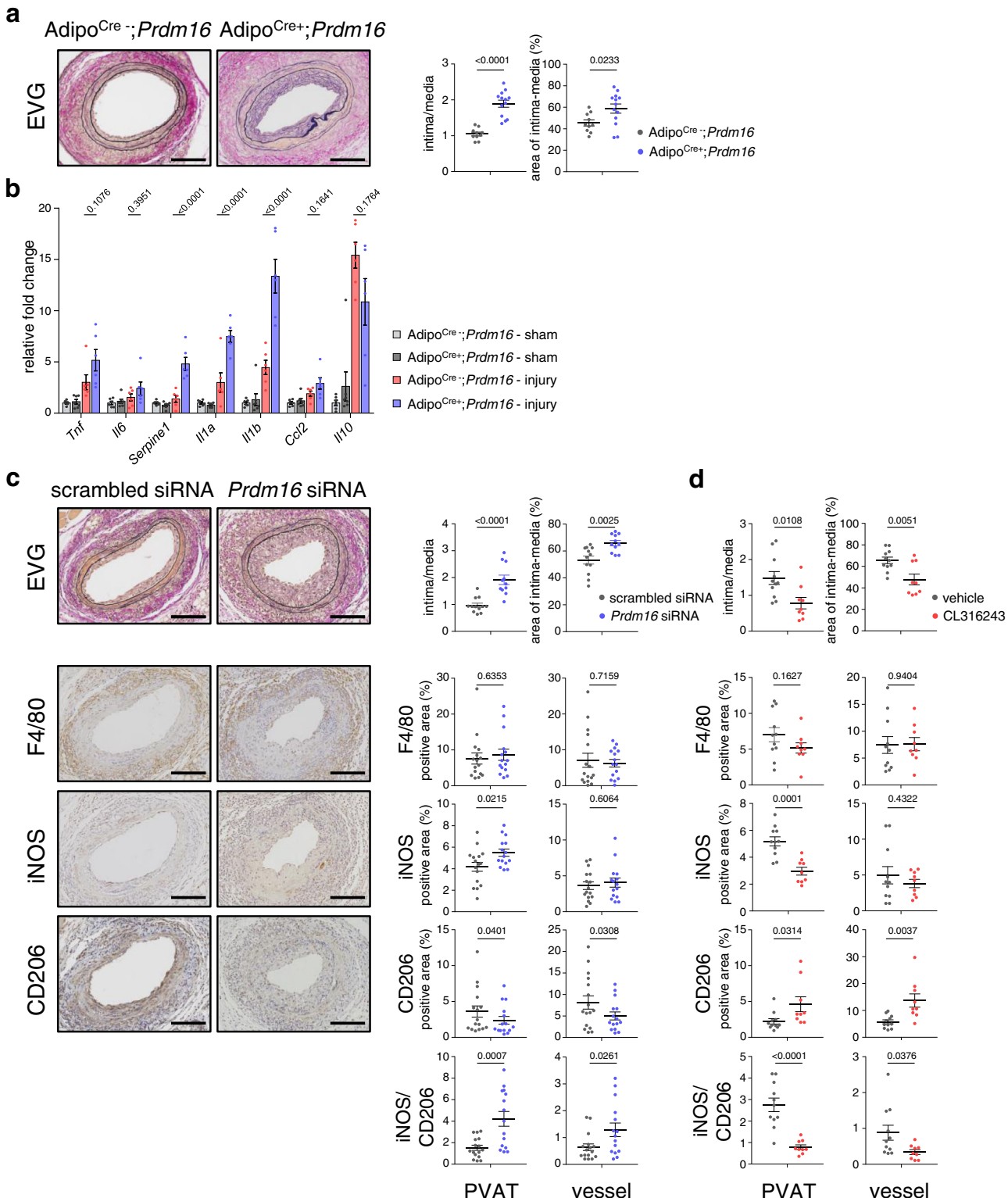

inhibits the growth of classically activated macrophages and induces phenotypic shift to an alternatively activated state, leading to accelerated resolution of macrophage inflammation, resulting in attenuation of pathological vascular remodeling after injury.

## PVAT beiging is detectable in human AAD

Acute aortic dissection (AAD) is the acute destruction of aortic wall that accompanies severe local inflammation[42]. We examined whether beiging was recognized in PVAT of human aorta with AAD. In addition

to the accumulation of macrophages, protein expression levels of markers for beige cells, such as UCP1 and CIDEA, and NRG4 were significantly increased likely as a compensatory response in the PVAT of AAD lesions compared with the PVAT without AAD (Fig. 5a, b and Supplementary Fig. 11a, b). To elucidate the role of PVAT beiging in AAD, we used an AAD murine model with the systemic administration of angiotensin II, β-aminopropionitrile, and N-nitro-L-arginine methyl ester[43–45]. CL316243 administration significantly reduced death due to dissection or rupture of the aorta (Fig. 5c), concomitant with the shift

**Fig. 2 | Modification of PVAT beiging alters vascular inflammatory response and remodeling after injury. a** EVG staining in FAs 14 days after injury of Adipo^Cre−^;Prdm16 mice and Adipo^Cre−^;Prdm16 (control) mice. The ratio of intima to media area (intima/media) and the % intima−medial area in the area surrounded by the external elastic lamina (area of intima−media) at 14 days after vascular injury were analyzed (Adipo^Cre−^;Prdm16, n = 10; Adipo^Cre+^;Prdm16, n = 13, unpaired two-tailed Student's t test). Scale bars represent 100 µm. **b** Gene expression of inflammatory cytokine markers in PVAT 14 days after vascular injury in Adipo^Cre+^;Prdm16 mice and Adipo^Cre−^;Prdm16 (control) mice (Adipo^Cre−^;Prdm16 − sham, n = 6; Adipo^Cre+^;Prdm16 − sham, n = 7; Adipo^Cre−^;Prdm16 − injury, n = 6; Adipo^Cre+^;Prdm16 − injury, n = 6, one-way ANOVA followed by Tukey−Kramer post-hoc test). **c, d** EVG and immunohistochemical staining for F4/80, iNOS (classically activated) and CD206 (alternatively activated) in FAs 14 days after injury in wild type mice treated with pluronic gel containing siRNA [Prdm16 or scrambled, (**c**)] or β3AR agonist [CL316243 or vehicle, (**d**)] applied to the PVAT surrounding the FA. The ratio of intima to media area (intima/media) and the % intima−medial area in the area surrounded by the external elastic lamina (area of intima−media) at 14 days after vascular injury were analyzed ((**c**) scrambled siRNA, n = 12; Prdm16 siRNA, n = 11, unpaired two-tailed Student's t test, (**d**) vehicle, n = 11; CL316243, n = 9, unpaired two-tailed Student's t test). The positive area of immunostaining and ratio of iNOS to CD206-positive area in arteries (vessel) or surrounding tissues (PVAT) were analyzed (**c** scrambled siRNA, n = 16; Prdm16 siRNA, n = 15, unpaired two-tailed Student's t test, **d** vehicle, n = 11; CL316243, n = 9, unpaired two-tailed Student's t test). Scale bars represent 100 µm. Data represent mean ± SEM. Source data are provided as a Source Data file.

of macrophage phenotypes in PVAT to the alternatively activated state evaluated by fluorescence-activated cell sorting (FACS) analysis (Fig. 5d, e and Supplementary Fig. 12). These findings suggest that beiging occurs in the PVAT of the human aorta during acute dissection and may regulate the inflammatory response during the development of AAD.

## Discussion

Vascular damage provokes regional inflammation and prolonged inflammation leads to pathological vascular remodeling[4,5]. The balance of pro- and anti-inflammation is critical for successful vascular wound healing and clinical outcomes[46,47]. The primary process of inflammation resolution is accomplished by the phenotypic conversion of pro-inflammatory macrophages into anti-inflammatory macrophages[47]. However, the mechanisms involved in this conversion have not yet been fully elucidated. This study revealed that endovascular injury induced 'beiging' of regional PVAT, and that inhibition of PVAT beiging exacerbated thickening of the intima after injury accompanied by an increased accumulation of pro-inflammatory macrophages, while activation of beiging conversely attenuated pathological vascular remodeling and accumulation of pro-inflammatory macrophages. These findings suggest that beiging of PVAT plays a pivotal role in changing the initial inflammation phase to its resolution after vascular injury. Various studies have suggested that endothelial dysfunction leads to the recruitment of circulating monocytes to the intima−medial layer[3,6]. In our study, the marked accumulation of monocytes was also identified in the outer tissues surrounding the vasculature from the early phase after vascular injury. The infiltrated monocyte-macrophages elicited PVAT beiging, resulting in the suppression of the excessive vascular inflammatory response to the injury. These data indicate a pivotal role of the "outside-in" manner in wound healing after vascular injury.

Local activation of PVAT beiging attenuated pathological vascular remodeling after injury, suggesting that PVAT acts directly on nearby arteries and mediates the pathophysiological response to vascular damage in a paracrine manner. We identified NRG4 as an anti-vascular remodeling factor derived from beige PVAT. NRG4 is a member of the NRG protein family, which acts via ErbB receptor tyrosine kinases. This molecule is highly expressed in the pancreas, skeletal muscles and BAT[48] and protects against diet-induced insulin resistance and hepatic steatosis through regulating hepatic lipogenic and cytoprotective signaling[38,40]. Recent studies have shown the expression of functional ErbB receptors on innate immune cells such as macrophages, dendritic cells and neutrophils[49], and the NRG4-ErbB4 axis exerts anti-inflammatory effects in macrophages by promoting apoptosis of classically activated macrophages but not alternatively activated macrophages[49,50]. Consistent with these findings, medium conditioned by PVAT-derived beige adipocytes significantly inhibited the number of classically activated macrophages, which was abolished by Nrg4 knockdown. Reduced levels of NRG4 have been reported to be associated with increased carotid intimal

thickness, increased angiographic severity of coronary artery disease and acute coronary syndrome[51−53]. In this study, PVAT beiging was detected in human aorta with acute dissection, suggesting that NRG4 secreted from beige PVAT might protect human aorta against AAD. The lineage of the beige cells that emerged in the PVAT was not examined in this study. Reportedly, smooth muscle cell has been considered a possible lineage of origin for PVAT[54], and Angueira et al. have shown an increase in Myh11+ smooth muscle cell-derived adipocytes in thoracic PVAT after treatment with rosiglitazone[55]. Since rosiglitazone reportedly recruits beige cells in fat tissues[56], the beige cells in PVAT may be originated from smooth muscle cells. Lineage tracing studies using inducible genetic techniques may provide further biological and functional insights into the beiging phenomenon in the PVAT. Our data showed that the beiging after injury was suppressed by the macrophage depletion or the local β3AR inhibition, implying that the PVAT beiging was elicited via β3AR signaling mediated by infiltrated macrophages. Meanwhile, the involvement of other signaling pathways of beiging such as succinate metabolisms and BMP4 in the beiging process in PVAT after injury remains to be elucidated[57,58]. In conclusion, the present study revealed that PVAT beiging plays a critical role in the vascular inflammatory response to injury by controlling macrophage inflammation and its resolution and suggests that PVAT beiging and NRG4 may be novel targets of vascular injury including AAD.

## Methods
### Animal studies

All animal procedures were approved by the University of Tokyo Ethics Committee for Animal Experiments and strictly adhered to the guidelines for animal experiments at the University of Tokyo. Male or, unless otherwise specified, ovariectomized female mice were used for the study. All wild type C57BL/6J mice were purchased from CLEA Japan. Adipo^Cre+^ mice (B6;FVB-Tg(Adipoq-Cre)1Evdr/J, #10803), Prdm16^flox/flox^ mice (B6.129-Prdm16^tm1.1Brsp^/J, #024992) and Ucp1^−/−^ mice (B6.129-Ucp1^tm1Kz^/J, #003124) were purchased from Jackson Laboratory. Conditional deletion of Prdm16 in adipocytes was achieved by crossing Prdm16^flox/flox^ homozygous mice with Adipo^Cre+^ hemizygous mice. Genotyping PCR was performed according to the protocol from Jackson Laboratory. All mice were maintained in specific pathogen-free conditions in the animal facilities of the University of Tokyo. They were housed in a controlled environment with a 12 h light/12 h dark cycle at a maintained temperature and kept with free access to food and water throughout the whole experiment period. Ambient room temperature was regulated at 73 ± 5 °F and humidity was controlled at 50 ± 10%. Endovascular injury of femoral arteries (FA) with wire insertion was performed at 10−12 weeks of age using a straight spring wire (0.38 mm in diameter, COOK) as previously described[59]. Arteries and outer tissues surrounding the vasculature were collected. Ovariectomies were performed in 8−10-week-old female mice 2 weeks before the endovascular injury, as previously described[60]. To selectively remove the macrophages, mice were given intraperitoneally clodronate liposomes

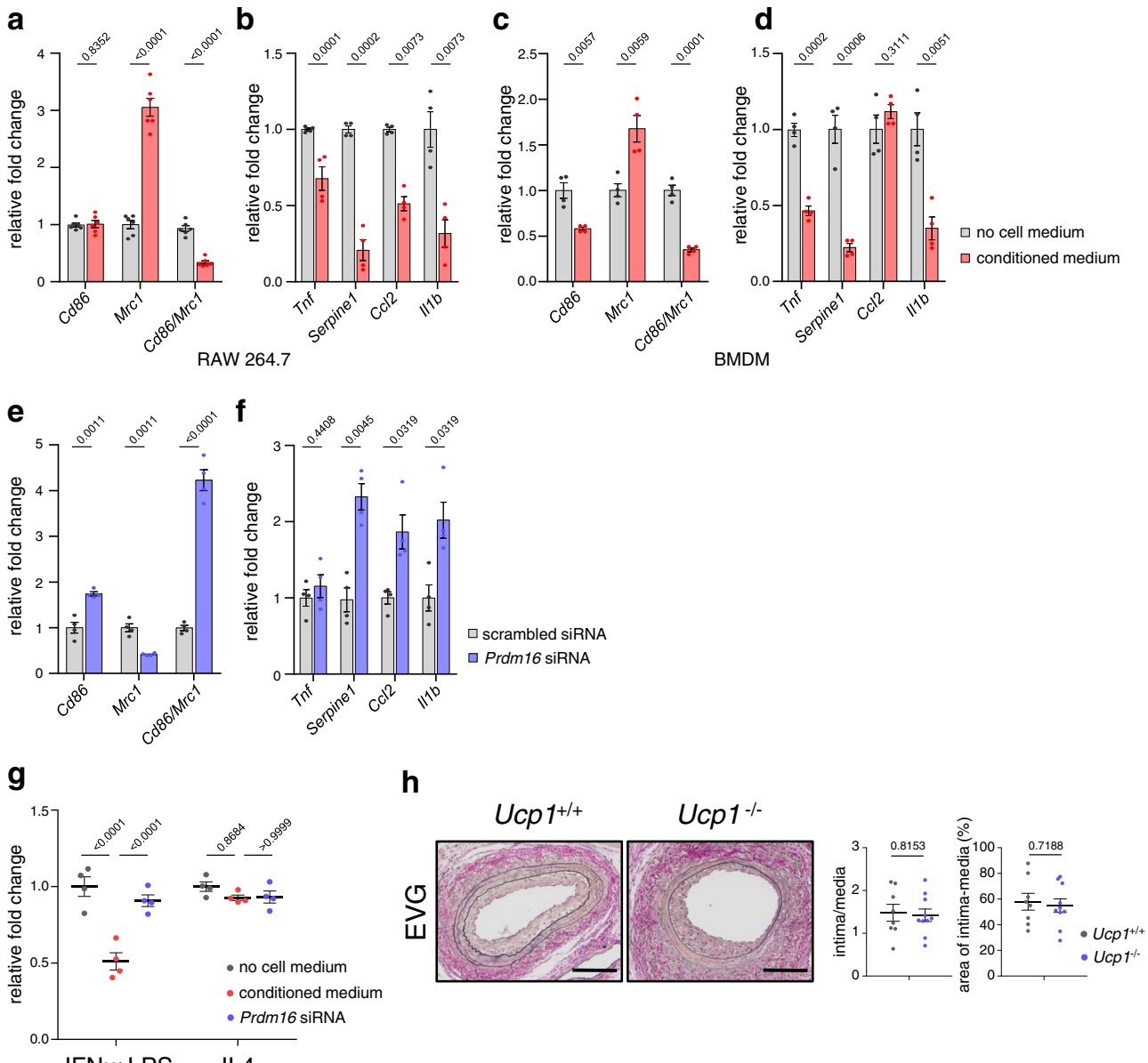

**Fig. 3 | Beige-differentiated PVAT shifts macrophage phenotypes to alternatively activated state. a–d** Cultured preadipocytes isolated from the PVAT stromal vascular fraction were differentiated into beige adipocytes, and media conditioned by the cells were added to RAW 264.7 macrophage cells (**a**, **b**) or bone marrow-derived monocytes (BMDMs) (**c**, **d**). After 48 h, mRNA was extracted from the RAW 264.7 cells or BMDMs and qRT-PCR was performed. The gene expression levels of macrophage phenotype markers [*Cd86* (classically activated), *Mrc1* (CD206, alternatively activated) and ratio of *Cd86* to *Mrc1*, (**a**, **c**)] and inflammatory cytokines [*Tnf*, *Serpine1*, *Ccl2*, and *Il1b*, (**b**, **d**)] were analyzed. Control cells were treated with media cultured in the absence of adipocytes (**a** $n = 6$ biological replicates, representative data of three different culture lines are shown, two-tailed $t$ tests with Holm-Sidak's correction for multiple comparisons, **b**–**d** $n = 4$ biological replicates, representative data of three different culture lines are shown, two-tailed $t$ tests with Holm-Sidak's correction for multiple comparisons. **e**–**f** PVAT-preadipocytes were introduced by siRNA of *Prdm16* or scrambled and stimulated with beige differentiation factors. Culture media conditioned by these cells were added to RAW 264.7 cells and mRNA was extracted 48 h after treatment. The results

of the qRT-PCR analyses of the gene expression levels of macrophage phenotype markers (**e**) and inflammatory cytokines (**f**) are shown ($n = 4$, biological replicates, representative data of three different culture lines are shown, two-tailed $t$ tests with Holm-Sidak's correction for multiple comparisons). **g** Cell growth analysis of RAW 264.7 cells 72 h after treatment with culture media conditioned by PVAT-derived beige adipocytes. RAW 264.7 cells were pre-treated with IFNγ (20 ng/ml) and LPS (20 ng/ml) or IL4 (10 ng/ml) for 24 h and treated with media cultured in the absence of adipocytes (gray dots), culture media conditioned by adipocytes (red dots) and culture media conditioned by adipocytes introduced by *Prdm16* siRNA (blue dots) ($n = 4$ biological replicates, representative data of three different culture lines are shown, one-way ANOVA followed by Tukey–Kramer post-hoc test). **h** EVG staining in the FA 14 days after injury in *Ucp1*[−/−] and wild type (*Ucp1*[+/+]) mice. Scale bars represent 100 μm. The ratio of intima to media area (intima/media) and % intima–medial area in the area surrounded by the external elastic lamina (area of intima–media) were analyzed (*Ucp1*[+/+], $n = 8$; *Ucp1*[−/−], $n = 10$, unpaired two-tailed Student's $t$ test). Data represent mean ± SEM. Source data are provided as a Source Data file.

(200 μl/mouse) or control phosphate-buffered saline (PBS) liposomes (Liposoma BV) on the day of the endovascular injury and 7 days after injury[61]. AAD was induced in mice by the simultaneous administration of angiotensin II (Ang II), β-aminopropionitrile (BAPN), and *N*-nitro-L-

arginine methyl ester (L-NAME)[43–45]. In the present study, AngII (1,000 ng/kg/min; Peptide Institute Inc.) and BAPN (150 mg/kg/day; Sigma-Aldrich) were administered simultaneously to 10-week-old male C57BL/6J mice using Alzet osmotic minipumps (Durect

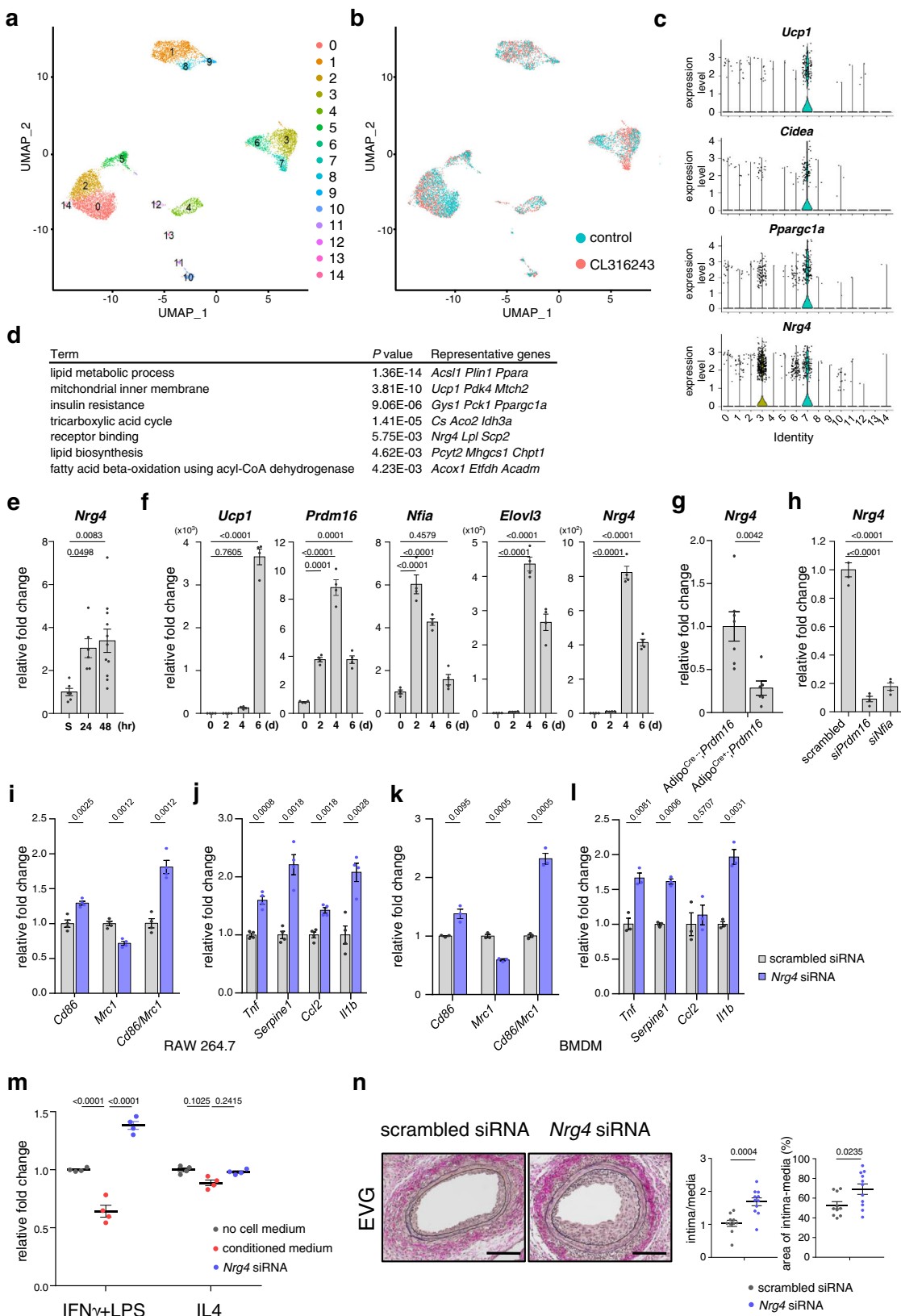

Corporation)[43–45]. L-NAME (100 mg/kg per day; Cayman Chemical Co.) was administered orally with drinking water at the same time as pump implantation until the end of the experiment[45]. For beiging activation studies in AAD murine model, β3AR agonist (CL316243, 1 mg/kg/day; Sigma-Aldrich) or vehicle PBS was administered daily for seven days using the Alzet osmotic minipump[62].

**Local administration of reagents**

Pluronic F-127 gel (Sigma-Aldrich) is a thermoreversible gel[63], which is liquid at refrigerated temperatures (4–5 °C), but gel upon warming to room temperature[64]. The gelation is reversible upon cooling. For local gene knockdown studies, 0.2 nmol/mouse Ambion in vivo pre-designed siRNA against *Prdm16* (s89042), *Nrg4* (s206928), or

**Fig. 4 | Identification of NRG4 as beige PVAT-derived secretory factor that controls macrophage phenotypes and growth. a, b** Clustering analysis of UMAP dimensionality reduction using scRNA-seq data in publicly available datasets of mouse iWAT treated with CL316243 or control (GSE 133486), dividing adipose tissue cells into 15 clusters which are color-coded by cell type (**a**) and by stimulation (CL316243 treatment [pink] or control [turquoise]) (**b**). **c** Violin plots of representative beige/BAT marker genes and *Nrg4* in each cluster indicated in **a. d** Gene ontology analysis for the top 100 regulated genes in C7 showing that several biological processes annotated by the genes were changed significantly. *Nrg4* was listed as a gene involved in receptor-binding. *P*-values are determined by Fisher's Exact test. **e** qRT-PCR analysis showing *Nrg4* gene expression in PVAT 24 and 48 h after injury (sham, *n* = 6; injury 24 h, *n* = 6; injury 48 h, *n* = 11, one-way ANOVA followed by Tukey–Kramer post-hoc test). **f** Time course of gene expressions in PVAT-derived preadipocytes stimulated by beige differentiation factors. Relative expression levels of beige/BAT markers and *Nrg4* are shown. (*n* = 4 biological replicates, representative data of three different culture lines are shown, one-way ANOVA followed by Tukey–Kramer post-hoc test). **g** Gene expression of *Nrg4* in PVAT 14 days after vascular injury in Adipo^Cre+^;*Prdm16* mice and Adipo^Cre−^;*Prdm16* (control) mice (Adipo^Cre−^;*Prdm16*, *n* = 7; Adipo^Cre+^;*Prdm16*, *n* = 6, unpaired two-tailed Student's *t* test). **h** qRT-PCR analysis showing *Nrg4* gene expression in PVAT-derived preadipocytes introduced by scrambled, *Prdm16* and *Nfia* siRNAs followed by stimulated with beige differentiation factors for 6 days (*n* = 4, biological replicates, representative data of three different culture lines are shown, one-way ANOVA

followed by Tukey–Kramer post-hoc test). **i–l** PVAT-derived preadipocytes were introduced by *Nrg4* or scrambled siRNA and stimulated with beige differentiation factors. Culture media conditioned by these cells were then added to RAW 264.7 cells (**i, j**) or BMDMs (**k, l**) and mRNA was extracted 48 h after treatment. The results of qRT-PCR for the gene expression levels of macrophage phenotype markers (**i, k**) and inflammatory cytokines (**j, l**) are shown ((**i, j**) *n* = 4 biological replicates, representative data of three different culture lines are shown, two-tailed *t* tests with Holm-Sidak's correction for multiple comparisons, (**k, l**) *n* = 3 biological replicates, representative data of three different culture lines are shown, two-tailed *t* tests with Holm-Sidak's correction for multiple comparisons). **m** Cell growth analysis of RAW 264.7 cells 72 h after treatment with conditioned media from PVAT-derived beige adipocytes. RAW 264.7 cells were pre-treated with IFNγ and LPS or IL4 for 24 h and treated with media cultured in the absence of adipocytes (gray dots), media conditioned by adipocytes (red dots) and media conditioned by adipocytes introduced by *Nrg4* siRNA (blue dots) (*n* = 4 biological replicates, representative data of three different culture lines are shown, one-way ANOVA followed by Tukey–Kramer post-hoc test). **n** EVG staining in FAs 14 days after injury in wild type mice treated with pluronic gel, containing siRNA against *Nrg4* or scrambled, that was applied to the PVAT surrounding the FA. The ratio of intima to media area (intima/media) and the % intima−medial area in the region surrounded by the external elastic lamina (area of intima−media) at 14 days after vascular injury were analyzed (scrambled siRNA, *n* = 10; *Nrg4* siRNA, *n* = 11, unpaired two-tailed Student's *t* test). Scale bars represent 100 µm. Data represent mean ± SEM. Source data are provided as a Source Data file.

negative control mixed with Invivofectamine 3.0 reagent (Thermo Fisher Scientific) was dissolved in pluronic gel on ice and administered locally in PVAT of the injured artery. For β3AR signal activation or inhibition studies, β3AR agonist (CL316243, 0.01 mg/kg; Sigma-Aldrich), β3AR antagonist (SR59230A, 0.02 mg/kg; Sigma-Aldrich), vehicle PBS, or vehicle dimethyl sulfoxide was dissolved in pluronic gel and administered in the PVAT.

## Histology and immunohistochemistry
Arteries including outer tissues fixed in Ufix (Sakura Finetek), were embedded in paraffin and sectioned into 4-µm-thick sections. Specimens were then stained by hematoxylin and eosin (H&E), and Elastica van Gieson (EVG) methods. The lumen, internal elastic lamina (IEL), and external elastic lamina (EEL) were defined, and the intimal (the tissue between lumen and IEL) and medial (tissue between IEL and EEL) areas were recorded using National Institutes of Health (NIH) ImageJ software. To evaluate vessel remodeling, the ratio of intima to media area (intima/media)[59] and the % intima−medial area = [(EEL area) − (luminal area)/EEL area] ×100 (%) (% area of intima−media)[65] were calculated as previously described. To evaluate adipocyte area, PVAT samples were stained with H&E and imaged with DP70 (Olympus). Uncompressed tif files were analyzed for adipocyte area by the Adiposoft plugin (v. 1.16) for NIH ImageJ[66,67]. Sections were immunohistochemically stained by using primary antibodies against F4/80 (Bio-Rad, MCA497GA, 1:400), UCP1 (Abcam, ab23841, 1:400), iNOS (Abcam, ab15323, 1:200), CD206 (Abcam, ab64693, 1:1600), CIDEA (Abnova, H00001149-M01, 1:50), or NRG4 (Thermo Fisher Scientific, PA5-96184, 1:1000) with Vectastain ABC kit (Vector Laboratories) according to the manufacturers' instructions. Peroxidase activity was visualized with DAB staining (Vector Laboratories), and sections were counterstained with hematoxylin. Images were acquired using the DP70 (Olympus) or BZ-X810 (KEYENCE) and analyzed using NIH ImageJ.

## Single-molecule fluorescence in situ hybridization
In situ hybridization was performed using the RNAscope Multiplex Fluorescent Reagent Kit v2 (Advanced Cell Diagnostics) according to manufacturers' instructions. The probes targeting mm-*Ucp1* (#455411) or mm-*Nrg4* (#493731, Advanced Cell Diagnostics) were hybridized, followed by RNAscope amplification and co-staining with fluorescein-conjugated WGA to detect cell borders. Slides were mounted with ProLong Diamond Antifade Mountant with DAPI (Life Technologies). Fluorescent signals were captured with the ×40 objective lens on a

laser scanning confocal microscope (LSM880, Zeiss). The number of positive dots was calculated using NIH ImageJ.

## PVAT-derived preadipocyte isolation
Eleven-week-old male wild type mice (for fluorescence-activated cell sorting analysis) or twelve-week-old female wild type mice (for cell culture) were sacrificed, and the PVAT from the thoracic aorta were extracted. PVAT was minced into small pieces using sharp round scissors and digested using gentleMACS (Miltenyi Biotec) and Adipose Tissue Dissociation Kit (Miltenyi Biotec), as previously described[68]. Digestion was performed in a MACSmix Tube Rotator (Miltenyi Biotec) at 37 °C for 40 min.

## Immortalization of PVAT-derived preadipocyte
A stromal vascular fraction of PVAT from 12-week-old female wild type mice was isolated and cultured in collagen-coated dishes, as previously described[68]. Preadipocytes isolated from the PVAT stromal vascular fraction were immortalized using the SV40 antigen as described previously[19].

## BMDM isolation
Bone marrow cells were flushed from the femurs and tibias of 10-week-old male wild type mice, and cultured at 37 °C in DMEM/F12 containing 1% streptomycin, 1% penicillin, and 10% fetal bovine serum (FBS) (Cosmo Bio) and in 40 ng/mL of recombinant mouse M-CSF (576406; BioLegend) for 7 days, as previously described[69]. The differentiated BMDMs were detached from plates using TrypLE (Thermo Fisher Scientific) and replated into 12-well tissue culture dishes at a density of 5 × 10^5 cells per well prior to cell stimulation.

## Cell culture
RAW 264.7 cells (European Collection of Authenticated Cell Culture), BMDMs, and PVAT-derived preadipocytes were cultured at 37 °C in DMEM/F12 with 1% streptomycin, 1% penicillin, and 10% FBS. For classically or alternatively activation, RAW 264.7 cells or BMDMs were treated with 20 µg/mL IFNγ (Miltenyi Biotec) + 20 µg/mL LPS (Sigma-Aldrich) or 10 µg/mL IL4 (Sigma-Aldrich) for 24 h, respectively, as previously described[70]. Recombinant murine NRG4 (100 ng/ml, Proteintech) or vehicle PBS was added to RAW 264.7 cells after classical activation. The growth of RAW 264.7 cells was measured with the Cell Titer Glo reagent (Promega) according to the manufacturer's instructions.

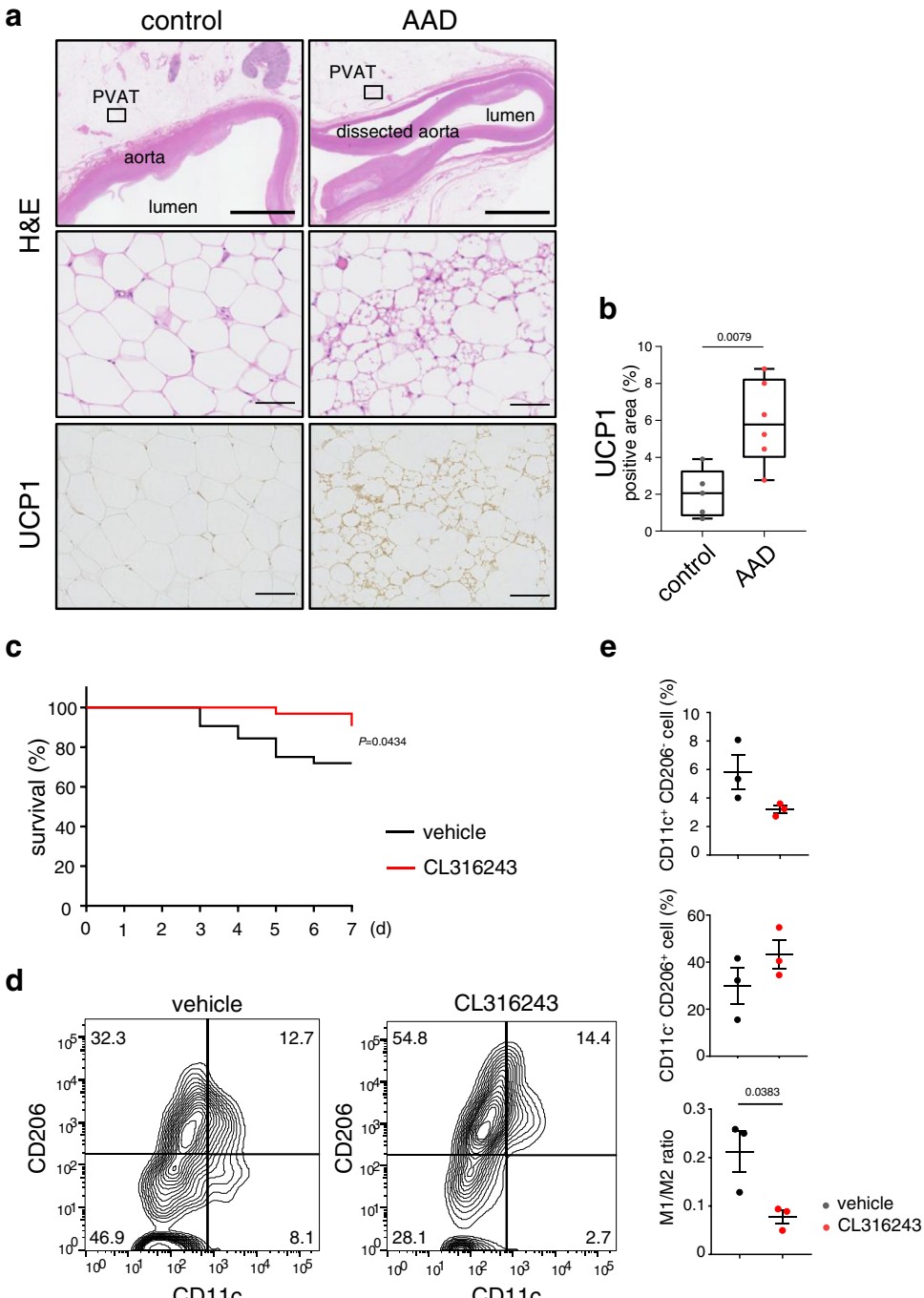

**Fig. 5 | PVAT beiging in human and murine aorta with acute dissection. a** Aorta tissue samples from patients with or without AAD were collected and H&E and immunohistochemical staining for UCP1 were performed. Scale bars represent 5 mm (thick bars) and 100 μm (thin bars). Images are representative of three independent experiments. **b** The % of UCP1⁺ areas in PVAT from patients with or without AAD were analyzed (control, $n=5$; AAD, $n=6$, two-sided Mann–Whitney $U$-test. Box plots denote the median, 25–75th percentiles, and minimum and maximum values). **c** Survival curves comparing vehicle-treated and CL316243-treated male AAD mice ($n=32$ for each group, Kaplan–Meyer analysis, followed by log-rank test). **d** Representative flow plots derived from the PVAT of AAD mice treated with CL316243 or vehicle. **e** Percentage of classically activated (M1; defined as CD11c⁺, CD206⁻) and alternatively activated (M2; defined as CD11c⁻, CD206⁺) macrophages, and the ratio of M1 to M2 macrophages measured by flow cytometric analysis are shown ($n=3$ for each group, unpaired two-tailed Student's $t$ test). Data represent the mean ± SEM. Source data are provided as a Source Data file.

## Differentiation of PVAT stromal vascular fraction to beige cells

PVAT-derived preadipocytes were differentiated to beige cells according to the previous reports[68]. Briefly, cells that reached confluence were treated for 48 h in medium containing 10% FBS, 850 nM insulin, 0.5 mM isobutylmethylxanthine, 125 nM indomethacin, 1 μM dexamethasone, 1 nM 3,3′,5-Triiodo-L-thyronine (T3) and 0.5 μM rosiglitazone. After 48 h treatment, the cell culture media were switched to media containing 10% FBS, 850 nM insulin, 1 nM T3 and 0.5 μM rosiglitazone and incubated for additional 5 days. Six to eight days after adding the induction medium, cells are fully differentiated to mature beige fat cells and are filled with oil droplets. Culture media conditioned by PVAT-derived beige adipocytes were added to RAW 264.7 cells, and mRNA expression of RAW 264.7 cells 24 h after stimulation was examined. For siRNA-mediated gene knockdown experiments in

cultured PVAT-derived adipocytes, siRNAs against *Prdm16*, *Nrg4*, *Erbb4* (s89042, s206928, s201303, respectively, Silencer select pre-designed siRNA, Thermo Fisher Scientific), *Nfia* (sc-36045, Santa Cruz Biotechnology) and negative control (Thermo Fisher Scientific) were transfected with Lipofectamine RNAiMAX reagent (Thermo Fisher Scientific) 2 days before confluence and following every 48 h, according to the manufacturer's instructions.

### Western blotting
Tissues were lysed in RIPA buffer containing 0.1% sodium dodecyl sulfate, 1% Triton X-100, 0.5% sodium deoxycholate, 150 mM NaCl, 50 mM Tris-Cl (pH 8.0), 1 mM EDTA supplemented with protease inhibitor (Roche). Proteins were separated by sodium dodecyl sulfate-polyacrylamide gel electrophoresis (SDS-PAGE), transferred to polyvinylidene difluoride membrane, and detected with the antibodies of anti-UCP1 (Abcam, ab23841, 1:1000) and anti-GAPDH (Cell Signaling Technology, #2118, 1:500).

### Quantitative reverse transcription-polymerase chain reaction
Total RNA from tissue samples was extracted using the RNeasy Mini Kit (Qiagen). Superscript VILO cDNA synthesis kit (Invitrogen) was used to generate cDNA. qRT-PCR was performed on QuantStudio 5 Real-Time PCR System (Thermo Fisher Scientific) using Thunderbird SYBR qPCR Mix (Toyobo). The specific primers are listed in Supplementary Table 1. Relative expression levels of target genes were calculated using the comparative CT method. Each sample was run in duplicate, and the results were systematically normalized using glyceraldehyde-3-phosphate dehydrogenase (*Gapdh*) for experiments using cultured cells or β-2 microglobulin (*B2m*) for experiments using mouse tissues.

### Bulk RNA-seq data processing
Outer tissues surrounding the FAs 48 h after injury or sham operation ($n = 2$ for each group) were collected. Total RNA (480 ng) from each sample was used to generate RNA-seq libraries using TruSeq Stranded mRNA Library Prep kit (Illumina). 150 bp paired-end sequencing reads were obtained using Illumina HiSeq 2500 platform. The Fastq sequences were initially checked for quality through the FastQC program v0.11.8. Adaptor sequences were removed by Cutadapt v1.18. Sequencing reads from RNA-seq libraries were then trimmed using Trimmomatic v0.39, where reads with alignment quality < Q33 were discarded, and aligned to the mouse reference genome (mm9) using STAR aligner v2.6.0a[71]. RSubread-2.0.1-FeatureCounts software for quantification and DESeq2 v.1.28.0 software[72] for differential analysis were used on R v.3.6.0.

### Single-cell RNA-seq data analysis
scRNA-seq data from publicly available datasets of mouse inguinal white adipose tissue (iWAT) nuclei treated with CL316243 (1 mg/kg/day for four days) or cold stimulation (4 °C for four days) (GSE 133486) were used. Among the original database that contains two types of scRNA-seq data (Adipo^Cre-^;*Il10ra*^flox/flox^ mice and Adipo^Cre+^;*Il10ra*^flox/flox^ mice)[36], the data from Adipo^Cre-^;*Il10ra*^flox/flox^ mice were used for the present analysis. The standard Seurat (V.4.0.1) for R (V.4.0.2) protocol ("FindIntegrationAnchors" and "IntegrateData") was used for the integration[73]. To observe the effects of CL316243 treatment, an alternative integration step was performed with Harmony (V.0.1)[74]. All sequenced cells were projected onto two dimensions using UMAP on R v.4.0.2. The optimal number of principal components used for UMAP dimensionality reduction was determined using the Jackstraw permutation approach and a grid-search of the parameters. Cells were divided into several clusters and color-coded according to cell types. Cells were also color-coded for the treatment group (CL316243 or cold stimulation) (pink) or control (turquoise). Marker genes were searched using the FindMarkers function in Seurat. Gene Ontology analysis was performed using Database for Annotation, Visualization, and Integrated Discovery ver. 6.8 (DAVID 6.8)[75]. Furthermore, the gene signature from the scRNA-seq data was used to deconvolute public bulk RNA-seq data to confirm the validity of the cell type clustering performed in this scRNA-seq re-analysis. We confirmed that the beige adipocytes also exist in iWAT treated with CL316243 in another public bulk RNA-seq study (GSE129083)[37] using MuSiC, a method for predicting cell abundance based on single-cell data[76].

### Fluorescence-activated cell sorting analysis
After the isolation of stromal vascular fraction of PVAT, cells were suspended in PBS containing 5% FBS. Zombie NIR Fixable Viability Kit (BioLegend) was used to label dead cells according to the manufacturer's instructions. Nonspecific binding of the antibodies to Fc receptors was blocked by using Mouse Fc receptor-blocking agent (BioLegend). Cells were incubated at room temperature 30 mins with FITC anti-mouse CD45 Antibody (BioLegend, #103107, 1:50), PE/Cyanine7 anti-mouse F4/80 antibody (BioLegend, #123113, 1:50), PE anti-mouse CD11b antibody (BioLegend, #101207, 1:50), APC anti-mouse CD206 antibody (BioLegend, #141707, 1:50), and Brilliant Violet 421 anti-mouse CD11c antibody (BioLegend, #117329, 1:50), then centrifuged and fixed with PBS containing 2% PFA. Cells were analyzed using a FACSMelody cell sorter (BD Biosciences). For the analysis of flow cytometry data, CD45$^+$, CD11b$^+$, and F4/80$^+$ cells were defined as macrophages, which could be further differentiated into classically (M1; defined as CD11c$^+$, CD206$^-$) and alternatively activated (M2; defined as CD11c$^-$, CD206$^+$) macrophages[77]. Data analyses were performed using FlowJo software (Tree Star).

### Statistics and reproducibility
Data were presented as means ± standard error of mean (SEM). All statistics are described in figure legends. In general, comparisons between two groups were performed using the two-tailed Student's *t* test or Mann–Whitney *U*-test, and multiple group comparisons were performed by one-way ANOVA followed by Tukey–Kramer post-hoc test. Survival curves were created using the Kaplan–Meier method and compared by a log-rank test. All statistical analyses were performed using the Prism software (GraphPad). *P*-values of <0.05 were considered statistically significant.

### Human studies
Pathological records were reviewed, and formalin-fixed paraffin-embedded descending aorta tissues were obtained from the archives of the Department of Pathology, The University of Tokyo Hospital. The aortic specimens of AAD were obtained by the autopsy of six patients who died of AAD, five males and one female, aged between 45 and 84 years; four of these patients had hypertension, one patient had suffered from a previous stroke, and three had a smoking history. The aortic specimens of controls were autopsy specimens from five patients who died of non-aortic causes, four males and one female, aged between 44 and 78 years; one patient had hypertension, one patient suffered from dyslipidemia, one with chronic kidney disease, one suffered from a previous stroke, and one had a smoking history. Patients with an apparent history of inherited aortic diseases such as Marfan syndrome were not included in the AAD cases and controls. The inclusion and exclusion criteria for patients with AAD were as follows: AAD cases were selected from consecutive pathological autopsies from 2006 to 2017 with a description of aortic dissection as a finding in autopsy report. Controls were selected from consecutive pathological autopsies from 2014 to 2017 with no significant changes in the aortic wall based on the autopsy. The immunostaining using primary antibodies against CD204 (Transgenic, KT022, 1:500) was performed using a Ventana Benchmark automated stainer (Ventana Medical Systems). The study followed the principles outlined in the Declaration of Helsinki. This study was approved by the ethics committee of the University of Tokyo (approval ID-2020019NI), and

written informed consent was waived because this is a retrospective study using existing tissue blocks. Instead, we used an opt out approach to provide participants with an informed choice about participation, although no patient in the cohort for screening used the opt out option.

## Reporting summary

Further information on research design is available in the Nature Research Reporting Summary linked to this article.

## Data availability

Source data are provided with this paper. The RNA-seq data have been deposited in GSE 206399, GSE133486, and GSE129083. The information of mouse genome (mm9) is available on UCSC Genome Browser (http://genome.ucsc.edu). The authors declare that all data are available in the article file and Supplementary information files, or available from the authors upon reasonable request. Source data are provided with this paper.

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

## Acknowledgements

We thank the participants for donating their tissues for this study; R. H. Karas for insightful discussions and suggestions; K. Akiba, Y. Ishiyama, Y. Nakashima, N. Matsuzaki, M. Oguchi, and T. Wada for the technical assistance; T. Kawahara for statistical consultation; and the Center for Disease Biology and Integrative Medicine, University of Tokyo, for mouse breeding and maintenance. This work was supported by the Japan Society for the Promotion of Science KAKENHI (18J13507, 20K22893, and 22K16097 to Y.A., 18K08096 and 21H02908 to K.U.), Japan Heart Foundation Research Grant, Japanese Circulation Society Clinical Research Grant, Japan Foundation for Applied Enzymology, SENSHIN Medical Research Foundation, Japan Arteriosclerosis Prevention Fund (to Y.A.), Japan Cardiovascular Research Foundation, Takeda Science Foundation, Mochida Memorial Foundation for Medical and

Pharmaceutical Research, Research Grants from Bristol Myers Squibb K.K., Tokyo Biochemical Research Foundation (to K.U.), JST FOREST Program (Grant Number 21466223), UTEC-UTokyo FSI Research Grant Program (to S.N.), and AMED (JP20ek0210152, JP18gm6210010, JP20ek0210141, JP20ek0109440, JP20ek0109487, JP17gm0810013, JP18km0405209, JP19ek0210118, JP19ek0109406, JP21ek0109543, JP21ek0109569, JP21tm0724601, JP22ama121016, JP22ek0210172, JP22ek0210167, JP22bm1123011) (to S.N. and I.K.).

## Author contributions

Y.A., K.U., and I.K. conceived the project, designed the study, and interpreted the results.; Y.A. and K.U. performed in vivo and in vitro experiments with the help of S.N., M.Katoh, M.Katagiri, M.Hashimoto, B.Z., G.N., A.O., Y.H., H.W., and E.T.; Y.A. and K.U. analyzed RNA-seq data under the supervision of S.N., K.I., and S.Y.; M.Hinata collected the human samples and clinical histories of the patients and helped immunohistochemistry under the supervision of T.U.; Y.A., K.U., and I.K. performed data analysis and wrote the manuscript under the supervision of S.N., K.I., N.T., H.M., T.U., T.Y., and E.T.; All authors contributed to the article and approved the submitted version.

## Competing interests

The authors declare no competing interests.
