## [Peer Review File · Nature Communications]

Reviewers' Comments:

Reviewer #1:

Remarks to the Author:

In the present study, Adachi et al. describe the infiltration of macrophages in PVAT during endothelial injury and the subsequent induction of beiging of PVAT. The authors demonstrate that PVAT plays pivotal roles in vascular inflammation and remodeling by the secretion of Nrg4 from beige PVAT. The paper is interesting and clearly written. The strengths of the manuscript are the multiple mouse models used and analyses that are performed. However, the novelty of the current manuscript is limited. BMP4 has recently been found to mediate browning of PVAT to prevent atherosclerosis through anti-inflammatory effects (PMID: 33895484). Specific concerns are outlined below.

Major points:

1. The ovariectomized female mice are generally used as the animal models for the menopause-related studies. Estrogens play considerable roles in blood vessels and lipid metabolism. The ovariectomized female mice might show much difference in vasculature compared with wild type female mice and thus the application of male mice is strongly suggested. Moreover, the aim of this study was not to investigate the effect of E2 on the accumulation of F4/80+ macrophages. The extended data Figures 1a and 1b are not needed.
2. Figure 2, Although previous studies have confirmed that Prdm16 regulates the browning or beiging of white adipose tissue, it is still necessary to verify beiging of PVAT by the staining of UCP1 in PVAT of Adipo-Cre+; Prdm16 mice and controls.
3. Figure 4, The scRNA-seq data showed that both C3 and C7 expressed Nrg4. Why did the authors consider Nrg4 from beige adipocyte (C7), but not from white adipocyte (C3), playing a predominant role?
4. The β 3-adrenergic receptor is widely expressed in adipose tissue and its function is very complex. The treatment with β 3-AR agonist CL316243 is not a classical WAT beiging model. Instead, CL316243 is often used to stimulate lipolysis. Cold stimulation is thus suggested to be used as a classical beiging model for further scRNA-seq.
5. There are various components of the conditioned medium from PVAT derived from beige adipocytes. To investigate the effects of NRG4 on macrophages (figure 3), purified NRG4 protein should be used to treat macrophages and its intracellular signal is also needed to be investigated.
6. Figure 3, It is suggested to use BMDM (bone marrow-derived monocyte) to further explore the function of NRG4.
7. Figure 5, Data from figure 5 did not support the protective role of PVAT beiging in AAD. PVAT in humans is mostly white adipose tissue, while it is a mixture of brown and white in rodents. Therefore, more evidence is needed in human AAD biopsies, such as the detection of macrophages and M1/M2 subclasses in PVAT from patients.

Minor points:

1. Page11, line242, NRG4 in brown adipose tissue is usually considered to regulate lipogenesis. Thus, the main function of NRG4 and its anti-inflammatory effects should be discussed in more detail.
2. The procedure of PVAT-derived preadipocyte isolation should be described in Methods.

Reviewer #2:

Remarks to the Author:

The proposed manuscript entitled "Beiging of perivascular adipose tissue regulates its inflammation and vascular remodeling" by Komuro and colleagues highlight the critical role of beige/Brite adipose tissue to regulate vascular injury induced inflammation and remodeling. The authors find that in response to femoral artery injury Ucp1+ beige adipocytes can be visualized after 48 hours. The authors show that the lack of beige fat reduces vascular repair and promotes inflammation. Moreover, they find that neuregulin 4 is produced by PVAT beige fat to stimulate macrophage polarity. In human samples, they find that beige fat is present in patients that have had acute aortic dissection. While this study demonstrates a new and potentially exciting area of vascular remodeling and beige adipose tissue crosstalk, the current data are limited and do not

appear to support the author's conclusions.

Comments:

1. Male vs female mouse models: The authors throughout the manuscript use ovariectomized female mice due to estradiol effects on vascular remodeling. However, removing or adding estradiol has been shown to influence adipose tissue beiging. For instance, Clegg and colleagues have shown that beiging is dependent upon estrogen receptor alpha activation (PMID: 30270132). Under the author's ovariectomized conditions, removing estrogen could dampen the PVAT beige fat induction. Similarly, adding back estrogen could also promote more beiging in PVAT than expected. It is unclear if the authors have controlled for these conditions, or have they noticed differences between female surgical sham or ovariectomized mice in beige PVAT injury? Clinically, males tend to develop vascular and cardiometabolic diseases at higher percentages than women. The authors should confirm that PVAT beige adipogenesis occurs in male vascular injury models and functions as they outlined in Figures 1-5.

1a (Minor critique). Ovariectomy is a surgical procedure that disrupts blood vessel integrity, creating injury, inflammation, and vascular remodeling. Can the authors determine that the surgery does not affect distal blood vessel PVAT beiging? That is; the authors are proposing that PVAT beige adipocytes are active endocrine cells promoting vascular remodeling and changes; thus, validating that the lack of beiging at distal locations would be a critical control.

2. The Ucp1 in situ hybridization and immunohistochemical staining are not convincing. The current images depict white adipocytes which are unilocular in morphology and are not multilocular, and appear to lowly express Ucp1 after injury compared to sham FA injury. New staining should be performed to convincingly demonstrate that beige adipocytes are present within the PVAT. Of note, the images shown in figure 5 clearly demonstrate the presence of multilocular adipocytes; however, the Ucp1 staining is considerably weak and does not suggest Ucp1 expression. Additionally, in figure 5, more gene expression profiling of beige and thermogenic markers should be performed.

3. Recent reports have shown that vascular smooth muscle cells can give rise to beige adipocytes as well as PVAT beige adipocytes (PMID: 24709624, PMID: 33846639, PMID: 26729601). Additionally, several reports have suggested that white adipocytes can interconvert into beige adipocytes (PMID: 23624403, PMID: 29657031). In the current manuscript, the cellular origins are unknown. This is critical since the authors are proposing that vascular injury induced beige adipogenesis; it would be critical to know if these cells are generated due to the injury and not due to the consequence of the injury. That is, do vascular smooth muscle cells that become disrupted in response to the injury generate beige adipocytes rather than facilitate vessel repair? Moreover, these newly generated cells may be masquerading as white adipocytes that are mobilizing lipids to assist in the repair of the vasculature and may not be "traditional" beige adipocytes. The authors should conduct simple lineage tracing experiments using several available inducible genetic tools (i.e. Pdgfra-CreERT2; Myh11-CreERT or Adiponectin-CreERT2 or rtTA) that can generate de novo beige adipocytes.

3a. The authors use CL316243 to induce beige adipogenesis. It is becoming clear that CL316243 induces beige adipocyte biogenesis primarily via interconverting white adipocytes into beige fat cells (PMID: 31540940, PMID: 29019320, PMID: 33294778). Thus, it is critical to understand how PVAT beige adipocytes are generated in response to injury. Are PVAT beige adipocytes generated the same way as CL316243 induced beige fat? This becomes critical when comparing scRNA sequencing data as pointed out in concern 7.

4. The effects observed on iNOS and CD206 using the Adiponectin-Prdm16 mouse model does not appear to be biologically significant even though they may be statistically significant. The authors should examine other markers of vascular inflammation including circulating marker such as P-selectin, interleukin-6, tumor necrosis factor- α , soluble intercellular adhesion molecule-1, and C-reactive protein. Moreover, the authors should conduct fluorescence activated cell sorting on the stromal vascular compartment to aid in inflammatory cell number and percentage quantification.

5 (minor critique). The authors claim that this effect on vascular remodeling is independent of Ucp1. Succinate metabolism has been considered to mediate thermogenesis independent of

Ucp1—might succinate fluxes account for the observed phenotype?

6. The authors need to validate macrophage switching in vivo using quantifying methods such as FACs. While the in vitro cultures provide preliminary evidence, in vivo identification is required.

7. Single cell RNA-sequencing: This reviewer is unclear why the author used pre-existing scRNA-seq datasets to arrive at neuregulin 4 upregulation. First, the authors are interested in PVAT but the scRNA sequencing was performed on subcutaneous inguinal WAT. Second, the authors performed bulk RNA sequencing on PVAT after injury, was neuregulin 4 upregulated in those datasets? The authors then show that neuregulin 4 is upregulated 48 hours after injury. Does it remain elevated throughout the injury like Ucp1 in Fig 1i? Is neuregulin 4 sourced from in vivo derived beige adipocytes? It is known that neuregulin 4 is secreted from thermogenic fat cells (PMID: 25401691); thus, it is not surprising that in vitro derived beige adipocytes secrete neuregulin 4. In vivo neuregulin 4 identification is strongly recommended. Third, the scRNA-sequencing was performed under CL316243 conditions; however, the authors are interested in vascular injury induced beige adipogenesis which may induce beige adipocytes differently or have altered beige adipocyte metabolomes etc. These differences need to be reconciled or not shown.

8. The authors demonstrate that in vitro derived PVAT induced beige adipocytes silenced against neuregulin 4 alters influence macrophage polarity. However, these data are insufficient and incomplete to determine if neuregulin 4 regulates macrophage polarity under these conditions. And these studies were performed using immortalized PVAT preadipocytes, not primary cells. The effects on gene expression are modest at best and again may not be biologically sufficient. Macrophage functional tests should be performed. The authors should inject mice with recombinant Nrg4 protein after vascular injury and assess macrophage polarity and number by flow cytometry and inflammatory and macrophage gene expression. In addition, the authors should use the pluronic gel (as done with Prdm16 Fig 2) with neuregulin 4 siRNA to demonstrate PVAT specificity in vivo. The best outcome would be to generate a genetic mouse model to alter Nrg4 expression to assess if beige adipocytes secrete Nrg4 to regulate vascular remodeling. But I understand this mouse model may be unattainable.

9. The AAD modeling in humans is lacking information to make formal conclusions based on the observed phenotype. First, are the samples from male or female adult humans? Second, what were the inclusion and exclusion criteria for the AAD patients (sex, age, disease states). AAD can be caused by several disease states or syndromes. It is not clear if the patients had hypertension, atherosclerosis, or other cardiovascular diseases, predisposing them to AAD. Third, were the biopsies collected at the time of AAD or follow-up? If these sample were collected at the time of AAD it would appear these data are contradictory to the authors hypothesis. The patients would be suffering from other cardiometabolic diseases predisposing them to vascular inflammation and the impinging aortic tear. Thus, based upon the data presented by the authors there should be less beige fat or functionally disrupted beige adipocytes, as its ability to protect the patients from AAD would be reduced? Therefore, based on the data concluded from the authors I would have expected less or dysfunctional beige adipose tissue in AAD patients.

10. These data hint at an interesting physiology under vascular remodeling and injury; however, it is unclear if the failure to induce beige adipocytes results in poor prognosis or development of vascular diseases. The authors need to identify a pathophysiological model to determine if beige adipose tissue influences vascular disease. Such suggested models include diet-induced obesity or ApoE-null mice.

11. The authors state that Ucp1 is involved in atherogenesis (ref 30, 32-35). However, their data suggest that Ucp1 does not mediate vascular remodeling after injury but beige fat—characterized by Ucp1 expression and function—is critical for injury remodeling. Again, the authors state that inflammation plays a critical role in the development of atherosclerosis and is one of the major reasons to initiate these studies. The authors should further discuss this topic to speculate how their data rederive differences between their study and published studies. Moreover, is neuregulin 4 secreted from beige adipocytes void of Ucp1?

Reviewer #3:

Remarks to the Author:

This new study from Adachi et al. examine the role of perivascular adipose tissue (PVAT) being in the regulation of vascular inflammation and remodeling. The present study utilizes a genetic mouse model in which adipose tissue cannot undergo "browning" (adipocyte Prdm16 knockout) in the setting of endovascular injury-induced inflammation. The data support a hypothesis that the macrophage-induced being of PVAT is needed to amount an eventual counterregulatory inflammatory response. The authors propose that the beige adipocyte derived NRG4 mediates this anti-inflammatory response by promoting the alternative anti-inflammatory phenotype of macrophages.

Overall, the paper is well written and the data are robust and important for the field. The paper nicely highlights a role for beige adipocytes in the resolution of injury-induced inflammation, involving a mechanism distinct from its classical thermogenic (UCP1-mediated) function. Despite these strengths, there are some weaknesses to current study that require attention.

1) Evidence that Nrg4 is a key regulatory molecule that mediates the effect of being in vivo is lacking. The authors are encouraged to knockout Nrg4 in vivo in a manner analogous to what was done for Prdm16 (e.g. siRNA). Also, the study of human AAD samples would be stronger if Nrg4 expression could be shown (e.g. histology).

2) A recent study by Seale et al. (Nature Metabolism 2021) provided a nice scRNA analysis of PVAT. The authors should discuss their findings with respect to this new literature. Some comparison of the datasets would be helpful.

Reviewer #4:

Remarks to the Author:

Adachi and Ueda et al reported that being of perivascular adipose tissue protected vascular injury through modulating inflammatory responses and vascular remodeling. To support this conclusion, the authors performed a number of analyses on the data collected from web lab-based experiments and public genomics data repository. Since my expertise is in bulk and single cell transcriptomics but not in adipogenesis, my comments will be largely restricted to this area. Overall, the bulk RNA-seq and single cell RNA-seq data present in this manuscript are not strong enough to support the main conclusion. Here are my comments for consideration.

1. The bulk RNA-seq data present in Figure 1e only contains n=2 from each group (sham versus injury). No statistics can be done on this low sample size so the data from this figure is not convincing. The authors are strongly advised to either increase the N or remove the figure.

2. The single nucleus RNA-seq data shown in Figure 4 and Extended Figures 3-4 were not generated by this study but from re-analysis of a public dataset (GSE133486). Several issues might arise from re-analysis of the data from other studies. First, the sample types are different. In the original paper, the snRNA-seq data were obtained from inguinal white adipose tissue (iWAT) and not PVAT. It is unclear if the adipocyte heterogeneity is different from tissue to tissue. Second, the authors only selected n=1 sample from each group to complete the analysis. This low sample size analysis, again, will compromise any conclusions drawn from it. Third, the authors identified a cluster from the analysis that only comes from the injury group. That would also happen when the integration algorithm cannot robustly correct for batch effect. The authors should apply multiple integrated pipelines (Seurat, Liger, Harmony, MNN, etc) on the same data and see if the cluster is from technical artifact of the integration or is a real cell type.

I strongly suggest that the authors should create their own single cell data from PVAT to demonstrate the transition of the adipocytes from normal to being state during vascular injury. However, if the single cell data is not critical in this study or other experimental data are sufficient to support the main conclusions, I have the following suggestions which might help strengthen the current analysis.

1. A number of bulk RNA-seq data (e.g. PMID:32139607 and PMID: 30116775) were published to study the beiging of white adipose tissue. The authors can use the gene signature from the single cell data to deconvolute these public bulk RNA-seq data to confirm that the beiging adipocyte also exists in the injured WAT from other studies. Computational algorithms for cell type deconvolution include MuSic, BSEQ-sc, Bisque, etc. This cross validation approach would make the data more convincing.

2. Two single cell RNA-seq datasets on beiging WAT from mouse (PMID: 29937373) and human (PMID: 32066997) are also publicly available. The authors can re-analyze these two datasets and see whether the same cell states are present in these two studies.

3. The author can perform ligand-receptor analysis (using CellPhoneDB, etc) on the single cell RNA-seq data to delineate the interactions between beiging adipocyte and immune cells or endothelial cells are mediated through Nrg4 and its associated receptors.

We thank the editor and the reviewers for their interest in our paper and thoughtful comments. We have addressed all comments in a point-by-point fashion as below. Based on the comments and suggestions, we have performed additional experiments and analyses and cleared up confusion over our previous text as well as expanded on topics requested by the reviewers. All changes within the manuscript are highlighted in blue. Overall, we believe that the revised manuscript has substantially improved with the clear and solid plot which is supported by more data than the original. We wish to thank all the reviewers for their attention to details and for their kind and informative comments.

Reviewer #1 (Remarks to the Author):

In the present study, Adachi et al. describe the infiltration of macrophages in PVAT during endovascular injury and the subsequent induction of beiging of PVAT. The authors demonstrate that PVAT plays pivotal roles in vascular inflammation and remodeling by the secretion of Nrg4 from beige PVAT. The paper is interesting and clearly written. The strengths of the manuscript are the multiple mouse models used and analyses that are performed. However, the novelty of the current manuscript is limited. BMP4 has recently been found to mediate browning of PVAT to prevent atherosclerosis through anti-inflammatory effects (PMID: 33895484). Specific concerns are outlined below.

Response: Thank you very much for the reviewer's helpful comments and suggestions. As the reviewer pointed out, the mechanisms of the PVAT browning remain to be elucidated. BMP4 reportedly induces the PVAT beiging and inhibits the chronic development of atherosclerosis¹. We revealed in this study that the beiging after injury was suppressed by the macrophage depletion (**Fig 1k, l**) or the local β 3AR inhibition (**Supplementary Fig. 3e**), implying that the PVAT beiging was elicited via β 3AR signaling mediated by infiltrated macrophages; however, other beiging regulators of PVAT, including BMP4, may affect the beiging process. We discussed this in the Discussion session (lines 310–315). The regulatory mechanisms of beige adipocytes in inflammation have been less understood. In this study, based on the comprehensive scRNA-seq data analysis, we newly identified NRG4 as a beige PVAT-derived anti-inflammatory factor that changes the polarity of macrophages in a paracrine manner, resulting in the proper resolution of inflammation after vascular injury. We believe that our findings unveil novel functions of PVAT beiging and NRG4 in vascular injury.

Comment 1. The ovariectomized female mice are generally used as the animal models for the menopause-related studies. Estrogens play considerable roles in blood vessels and lipid metabolism. The ovariectomized female mice might show much difference in vasculature compared with wild type female mice and thus the application of male mice is strongly suggested. Moreover, the aim of this study was not to investigate the effect of E2 on the accumulation of F4/80+ macrophages. The extended data Figures 1a and 1b are not needed.

Response 1: As suggested by the reviewer, we have performed a body of experiments using male mice. Consistent with the ovariectomized female mice, we confirmed that PVAT beiging after vascular injury also occurred in male mice by UCP1 immunostaining and calculation of the adipocyte size (**Supplementary Fig. 1c, d**). Vascular injury induced more prominent remodeling in Adipo^{Cre+};Prdm16 male mice than in Adipo^{Cre-};Prdm16 male mice at 14 days after injury (**Supplementary Fig. 2c**). Regarding the effect of estrogen on beiging, the expression of beige markers was not affected by the presence or absence of ovariectomy in the sham PVAT (**Supplementary Fig. 1b**). In injured PVAT, the gene expression levels of beige markers, such as *Ucp1* and *Elovl3*, was significantly upregulated in ovariectomized mice compared to that in non-ovariectomized mice (**Supplementary Fig. 1b**). Although further studies are needed to elucidate the mechanisms by which ovariectomy increases the beiging response to vascular injury, estrogen may inhibit beiging after injury by repressing macrophage accumulation in the PVAT, which induces beiging. As the reviewer pointed out, investigating the effect of E2 on the accumulation of F4/80⁺ macrophages was not the aim of this study; thus, **Extended Data Figure 1a, b** in the initial manuscript have been deleted.

Comment 2. Figure 2, Although previous studies have confirmed that Prdm16 regulates the browning or beiging of white adipose tissue, it is still necessary to verify beiging of PVAT by the staining of UCP1 in PVAT of Adipo-Cre+; Prdm16 mice and controls.

Response 2: Along with the reviewer's comment, we performed UCP1 staining in Adipo-Cre⁺;Prdm16 and Adipo-Cre⁻;Prdm16 control mice, and confirmed that beiging was suppressed in PVAT after vascular injury in Adipo-Cre⁺;Prdm16 mice (**Supplementary Fig. 2b**).

Comment 3. Figure 4, The scRNA-seq data showed that both C3 and C7 expressed Nrg4. Why did the authors consider Nrg4 from beige adipocyte (C7), but not from white adipocyte (C3), playing a predominant role?

Response 3: Cluster analysis using single cell RNA-seq using inguinal WAT revealed C7 mostly consisted of β 3-AR agonist CL316243-treated cells (**Fig. 4a, b, Supplementary Fig.**

4a). C7 represents beige adipocytes induced by β 3-AR activation, and gene ontology analysis for the significantly regulated genes in C7 identified *Nrg4* as a highly expressed secretory factor involved in the molecular function of receptor-binding (**Fig. 4c, d**). Although *Nrg4* was expressed in the white adipocyte cluster C3, the upregulation of *Nrg4* was significantly attenuated in *Adipo^{Cre+};Prdm16* mice and PVAT-derived preadipocytes transduced with siRNA against *Prdm16* or *Nfia* (**Fig. 4g, h**). The knockdown of *Nrg4* in perivascular tissue exacerbated vascular remodeling after injury (**Figure 4n**). Taken together, these results suggest that beige adipocytes are the primary source of *Nrg4* upregulated in the PVAT after vascular injury and that the *Nrg4* upregulation in beige PVAT after injury is functionally relevant in the regulation of the inflammatory response and subsequent vascular remodeling.

Comment 4. The β 3-adrenergic receptor is widely expressed in adipose tissue and its function is very complex. The treatment with β 3-AR agonist CL316243 is not a classical WAT beiging model. Instead, CL316243 is often used to stimulate lipolysis. Cold stimulation is thus suggested to be used as a classical beiging model for further scRNA-seq.

Response 4: We appreciate the reviewer's thoughtful comments. We accordingly re-analyzed publicly available scRNA-seq data from murine inguinal WAT treated with cold stimulation. **Supplementary Fig. 8a, b** show the clustering analysis of UMAP dimensionality reduction using scRNA-seq data in publicly available datasets of mouse inguinal WAT treated with cold stimulation or control (GSE 133486)², dividing adipose tissue cells into 12 clusters color-coded by cell type (**Supplementary Fig. 8a**) and stimulation (cold stimulation [pink] or control [turquoise]) (**Supplementary Fig. 8b**). Cluster 9 (C9) cells predominantly composed of cold-stimulated cells and abundantly expressed beige/BAT marker genes and *Nrg4* (**Supplementary Fig. 8c**). The gene ontology analysis revealed that *Nrg4* was identified as an upregulated gene belonging to the molecular function of receptor binding (**Supplementary Fig. 8d**). These results suggest that *Nrg4* is upregulated in beige adipocytes induced not only by a β 3-AR agonist CL316243 but also by cold stimulation.

Comment 5. There are various components of the conditioned medium from PVAT derived from beige adipocytes. To investigate the effects of NRG4 on macrophages (figure 3), purified NRG4 protein should be used to treat macrophages and its intracellular signal is also needed to be investigated.

Response 5: As suggested by the reviewer, to investigate the effects of NRG4 on macrophages,

we performed *in vitro* experiments using purified murine recombinant NRG4 protein. NRG4 protein decreased the *Cd86/Mrc1* ratio and mRNA expression levels of inflammatory cytokines in macrophages (**Supplementary Fig. 10e, f**). In addition, the knockdown of Erb-B2 receptor tyrosine kinase 4 (ErbB4), a receptor of NRG4, in macrophages abolished alternative activation and anti-inflammatory effects of culture media conditioned by PVAT-derived beige adipocytes (**Supplementary Fig. 10g–i**), suggesting the importance of intracellular signaling mediated by ErbB4 in interpreting the anti-inflammatory effects of *Nrg4*.

Comment 6. Figure 3, It is suggested to use BMDM (bone marrow-derived monocyte) to further explore the function of NRG4.

Response 6: According to the reviewer's suggestion, the functional analysis of NRG4 was performed using BMDM in addition to RAW 264.7 macrophages. Culture media conditioned by PVAT-derived beige adipocytes induced phenotypic changes from classical to alternative activation and the downregulation of inflammatory genes in BMDM (**Fig. 3c, d**). These changes were abolished in culture media conditioned by PVAT-derived beige adipocytes introduced with siRNA for *Nrg4* (**Fig. 4k, l**), suggesting that NRG4 is critically involved in the beige adipocyte-induced resolution of macrophage inflammation in BMDM.

Comment 7. Figure 5, Data from figure 5 did not support the protective role of PVAT being in AAD. PVAT in humans is mostly white adipose tissue, while it is a mixture of brown and white in rodents. Therefore, more evidence is needed in human AAD biopsies, such as the detection of macrophages and M1/M2 subclasses in PVAT from patients.

Response 7: As suggested by the reviewer, we performed immunohistochemical staining for macrophages in PVAT from patients and observed that the expression of iNOS, an M1 macrophage marker, was significantly increased in PVAT of AAD patients (**Supplementary Fig. 11b [top]**). Furthermore, we found that the expression of Cd204, an M2 macrophage marker, was also increased in PVAT of AAD patients, although it did not reach the level of statistical significance ($p=0.0823$, **Supplementary Fig. 11b [bottom]**). As mentioned by the reviewer, the protective role of PVAT being cannot be determined solely from immunohistochemical analyses of pathological specimens from human patients with AAD, and we thus used an AAD murine model to elucidate the protective role of PVAT being in AAD. CL316243 administration significantly reduced deaths due to dissection or rupture of the aorta (**Fig. 5c**), concomitant with the shift of macrophage phenotypes in PVAT into the alternatively activated state evaluated by fluorescence-activated cell sorting analysis (**Fig. 5d, e** and

Supplementary Fig. 12). These findings suggest that beiging occurs in the PVAT of the human aorta during acute dissection and may regulate the inflammatory response during the development of AAD.

Comment 8. Page 11, line 242, NRG4 in brown adipose tissue is usually considered to regulate lipogenesis. Thus, the main function of NRG4 and its anti-inflammatory effects should be discussed in more detail.

Response 8: As suggested by the reviewer, we mentioned the general function of NRG4 and its anti-inflammatory action in more detail in the Discussion section as follows: NRG4 is a member of the NRG protein family, which acts via ErbB receptor tyrosine kinases. This molecule is highly expressed in the pancreas, skeletal muscles and brown adipose tissue³ and protects against diet-induced insulin resistance and hepatic steatosis through regulating hepatic lipogenic and cytoprotective signaling^{4,5}. Recent studies have shown the expression of functional ErbB receptors on innate immune cells such as macrophages, dendritic cells and neutrophils⁶, and the NRG4-ErbB4 axis exerts anti-inflammatory effects in macrophages by promoting apoptosis of classically activated macrophages but not alternatively activated macrophages^{6,7} (lines 289–297).

Comment 9. The procedure of PVAT-derived preadipocyte isolation should be described in Methods.

Response 9: We have described a detailed procedure for PVAT-derived preadipocyte isolation in the Methods section (lines 389–396).

Reviewer #2 (Remarks to the Author):

The proposed manuscript entitled “Beiging of perivascular adipose tissue regulates its inflammation and vascular remodeling” by Komuro and colleagues highlight the critical role of beige/Brite adipose tissue to regulate vascular injury induced inflammation and remodeling. The authors find that in response to femoral artery injury Ucp1+ beige adipocytes can be visualized after 48 hours. The authors show that the lack of beige fat reduces vascular repair and promotes inflammation. Moreover, they find that neuregulin 4 is produced by PVAT beige fat to stimulate macrophage polarity. In human samples, they find that beige fat is present in patients that have had acute aortic dissection. While this study demonstrates a new and potentially exciting area of vascular remodeling and beige adipose tissue crosstalk, the current data are limited and do not appear to support the author’s conclusions.

Comment 1. Male vs female mouse models: The authors throughout the manuscript use ovariectomized female mice due to estradiol effects on vascular remodeling. However, removing or adding estradiol has been shown to influence adipose tissue beiging. For instance, Clegg and colleagues have shown that beiging is dependent upon estrogen receptor alpha activation (PMID: 30270132). Under the author’s ovariectomized conditions, removing estrogen could dampen the PVAT beige fat induction. Similarly, adding back estrogen could also promote more beiging in PVAT than expected. It is unclear if the authors have controlled for these conditions, or have they noticed differences between female surgical sham or ovariectomized mice in beige PVAT injury? Clinically, males tend to develop vascular and cardiometabolic diseases at higher percentages than women. The authors should confirm that PVAT beige adipogenesis occurs in male vascular injury models and functions as they outlined in Figures 1-5.

Response 1: We thank the reviewer for the insightful comment. We confirmed that PVAT beiging after vascular injury also occurred in male mice by UCP1 immunostaining and calculation of the adipocyte size (**Supplementary Fig. 1c, d**). Vascular injury induced more prominent remodeling in Adipo^{Cre+};Prdm16 male mice than in Adipo^{Cre-};Prdm16 male mice at 14 days after injury (**Supplementary Fig. 2c**). Regarding the effect of estrogen on beiging, the expression of beige markers was not affected by the presence or absence of ovariectomy in the sham PVAT (**Supplementary Fig. 1b**). In injured PVAT, the gene expression of beige markers, such as *Ucp1* and *Elovl3*, was significantly upregulated in ovariectomized mice compared to that in non-ovariectomized mice (**Supplementary Fig. 1b**). Although further studies are needed to elucidate the mechanisms by which ovariectomy increases the response to vascular

injury, estrogen may inhibit beiging after injury by repressing macrophage accumulation in the PVAT, which induces beiging. Additionally, because PVAT beiging may change depending on the menstrual cycle, we attempted to stabilize the extent of PVAT beiging by ovariectomy when female mice were used in this study.

Comment 1a (Minor critique). Ovariectomy is a surgical procedure that disrupts blood vessel integrity, creating injury, inflammation, and vascular remodeling. Can the authors determine that the surgery does not affect distal blood vessel PVAT beiging? That is; the authors are proposing that PVAT beige adipocytes are active endocrine cells promoting vascular remodeling and changes; thus, validating that the lack of beiging at distal locations would be a critical control.

Response 1a: To assess the effects of ovariectomy on PVAT beiging, we compared the expression levels of BAT markers such as *Ucp1* and *Elovl3* between mice with ovariectomy and those with sham-operated intact ovary mice, and found that ovariectomy did not affect PVAT beiging when femoral arteries were not injured (**Supplementary Fig. 1b**).

Comment 2. The Ucp1 in situ hybridization and immunohistochemical staining are not convincing. The current images depict white adipocytes which are unilocular in morphology and are not multilocular, and appear to lowly express Ucp1 after injury compared to sham FA injury. New staining should be performed to convincingly demonstrate that beige adipocytes are present within the PVAT. Of note, the images shown in figure 5 clearly demonstrate the presence of multilocular adipocytes; however, the Ucp1 staining is considerably weak and does not suggest Ucp1 expression. Additionally, in figure 5, more gene expression profiling of beige and thermogenic markers should be performed.

Response 2: According to the reviewer's suggestion, we improved the quality of the images for UCP1 immunohistochemistry and *in situ* hybridization (**Figure 1f, g and 5a**), where smaller and multilocular adipocytes were observed in PVAT after vascular injury than those after sham injury (**Figure 1g-i**). Because the human aortic specimen in this study was a formalin-fixed autopsy specimen and not a biopsy specimen, gene expression evaluation by qPCR could not be performed. Instead, we performed immunostaining and confirmed that the expression of CIDEA, another known beige marker, was significantly upregulated in the PVAT of AAD patients compared to controls (**Supplementary Fig. 11a** [top]). These results support our conclusions that beiging in PVAT is elicited by acute vascular injury. We added these observations in the revised manuscript.

*Comment 3. Recent reports have shown that vascular smooth muscle cells can give rise to beige adipocytes as well as PVAT beige adipocytes (PMID: 24709624, PMID: 33846639, PMID: 26729601). Additionally, several reports have suggested that white adipocytes can interconvert into beige adipocytes (PMID: 23624403, PMID: 29657031). In the current manuscript, the cellular origins are unknown. This is critical since the authors are proposing vascular injury induced beige adipogenesis; it would be critical to know if these cells are generated due to the injury and not due to the consequence of the injury. That is, do vascular smooth muscle cells that become disrupted in response to the injury generate beige adipocytes rather than facilitate vessel repair? Moreover, these newly generated beige cells may be masquerading as white adipocytes that are mobilizing lipids to assist in the repair of the vasculature and may not be “traditional” beige adipocytes. The authors should conduct simple lineage tracing experiments using several available inducible genetic tools (i.e. *Pdgfra-CreERT2*; *Myh11-CreERT* or *Adiponectin-CreERT2* or *rtTA*) that can generate de novo beige adipocytes.*

Response 3: Thank you for the insightful comments. As pointed out by the reviewer, previous reports have suggested multiple possibilities for the origin of the beige adipocytes, including smooth muscle cells and white adipocytes⁸⁻¹¹. Regarding the origin for PVAT, smooth muscle cell has been reported as a promising lineage^{12,13}. Chang L. et al have reported a PVAT hypoplasia in a mouse model deficient in peroxisome proliferator-activated receptor- γ in SM22⁺ smooth muscle cells¹². Consistently, Angueira AR *et al.* have reported an increase in MYH11⁺ smooth muscle cell-derived adipocytes in PVAT after treatment with rosiglitazone¹³. Since rosiglitazone reportedly recruits beige cells in fat tissues¹⁴, the beige cells in PVAT may be originated from smooth muscle cells. To address this matter, we developed *Tagln* (Transgelin, coding SM22)-*Cre*^{+/-};*Prdm16*^{flx/flx} mice, referred to as *Tagln*^{Cre+};*Prdm16* mice, in which beiging is inhibited in SM22-positive smooth muscle cell lineage. PVAT beiging was significantly attenuated in PVAT surrounding thoracic aorta in *Tagln*^{Cre+};*Prdm16*^{flx/flx} mice compared with *Tagln*^{Cre-};*Prdm16*^{flx/flx} control mice (**Figure 1. for reviewer #2**). This observation indicated that beiging cells in PVAT belong to a lineage of SM22-positive smooth muscle cells at least in the thoracic aorta. We would like to examine the lineage of PVAT beiging cells in future studies and, in the revised manuscript, have included the following limitations to the Discussion section: “The lineage of the beige cells that emerged in the PVAT was not examined in this study. Reportedly, smooth muscle cell has been considered a possible lineage of origin for PVAT¹², and Angueira AR *et al.* have shown an increase in *Myh11*⁺ smooth

muscle cell-derived adipocytes in thoracic PVAT after treatment with rosiglitazone¹³. Since rosiglitazone reportedly recruits beige cells in fat tissues¹⁴, the beige cells in PVAT may be originated from smooth muscle cells. Lineage tracing studies using inducible genetic techniques may provide further biological and functional insights into the beiging phenomenon in the PVAT.” (lines 303–310).

Figure 1. for reviewer #2: Gene expression levels of BAT (*Ucp1* and *Prdm16*) and WAT [*Cfd* (Adipsin) and *Rstn* (Resistin)] markers in PVAT surrounding thoracic aorta in *Tagln^{Cre+};Prdm16* mice compared with *Tagln^{Cre-};Prdm16* control littermates (n=3–5 for each group, multiple *t*-tests with Holm-Sidak's correction). Data represent mean ± SEM **P* < 0.05, ***P* < 0.01.

Comment 3a. The authors use CL316243 to induce beige adipogenesis. It is becoming clear that CL316243 induces beige adipocyte biogenesis primarily via interconverting white adipocytes into beige fat cells (PMID: 31540940, PMID: 29019320, PMID: 33294778). Thus, it is critical to understand how PVAT beige adipocytes are generated in response to injury. Are PVAT beige adipocytes generated the same way as CL316243 induced beige fat? This becomes critical when comparing scRNA sequencing data as pointed out in concern 7.

Response 3a: We assume that beige adipocytes in PVAT are generated by the β3AR signaling activation, the similar way to CL316243. M2 macrophages have been reported to cause beiging through the β3AR signaling activation in direct or indirect ways. Directly, M2 macrophages themselves secrete catecholamines that promote WAT beiging¹⁵. Indirectly, M2 macrophages modulate the sympathetic innervation to adipose tissues through the secretion of *Slit3* that activates PKA signaling in the sympathetic nerves¹⁶. In our study, the results of bulk RNA-seq and CD206 immunostaining showed that M2 macrophages accumulated in the PVAT from an early stage after injury (**Supplementary Fig. 1a** and **Supplementary Fig. 3a, b**). We also

revealed that the beiging after injury was suppressed by the macrophage depletion (**Fig. 1k, l**), and that the beiging in PVAT elicited by vascular injury was attenuated by a β 3AR antagonist (SR59230A) (**Supplementary Fig. 3e**). The conditioned media from RAW 264.7 polarized to M2 by interleukin-4 upregulated BAT markers in PVAT-derived preadipocytes (**Figure 2. for reviewer #2**). These results support our hypothesis that beige adipocytes in PVAT are generated by the β 3AR signaling activation. Meanwhile, other beiging regulators of PVAT, such as succinate metabolisms and BMP4 that reportedly induce the PVAT beiging and inhibit the chronic development of atherosclerosis^{1,17}, may affect the beiging process in PVAT after injury. We accordingly modified our manuscript (lines 310–315).

Figure 2. for reviewer #2: qRT-PCR analysis showing BAT marker (*Ucp1* and *Elovl3*) gene expression levels in PVAT-derived preadipocytes treated with conditioned media from RAW 264.7 macrophages stimulated with interleukin 4 (IL4) or vehicle control (n=8 biological replicates, representative data of three different culture lines are shown, multiple *t*-tests with Holm-Sidak's correction). Data represent mean \pm SEM ****P* < 0.001.

Comment 4. The effects observed on iNOS and CD206 using the Adiponectin-Prdm16 mouse model does not appear to be biologically significant even though they may be statistically significant. The authors should examine other markers of vascular inflammation including circulating marker such as P-selectin, interleukin-6, tumor necrosis factor- α , soluble intercellular adhesion molecule-1, and C-reactive protein. Moreover, the authors should conduct fluorescence activated cell sorting on the stromal vascular compartment to aid in inflammatory cell number and percentage quantification.

Response 4: We appreciate the reviewer's insightful advice. Since systemic inflammatory response by analyzing the circulating inflammatory markers may not directly reflect the vascular specific inflammation, we additionally conducted qRT-PCR analysis in PVAT 14 days after vascular injury of Adipo^{Cre+};Prdm16 mice and Adipo^{Cre-};Prdm16 (control) mice to

estimate the biological inflammatory response. As shown in **Figure 2b**, inflammatory cytokine markers, such as *Serpine1*, *Il1a* and *Il1b*, were significantly upregulated in the PVAT of *Adipo^{Cre+};Prdm16* mice, suggesting upregulation of biological inflammation in the PVAT of *Adipo^{Cre+};Prdm16* mice. For fluorescence-activated cell sorting (FACS) analysis, we collected femoral PVAT from ten mice to extract the stromal vascular fraction (SVF); however, the femoral PVAT samples were very small (0.03-0.1 mg/mouse), and it was technically difficult to extract a sufficient amount of SVF, even when we used ten samples. Therefore, we used aortic PVAT with drug-induced vascular injury (simultaneous administration of angiotensin II, β -aminopropionitrile, and N-nitro-L-arginine methyl ester), which is known to induce aortic dissection in mice, to examine macrophage polarization using FACS after CL316243 treatment to promote beiging. The results showed that the administration of CL316243 shifted the PVAT macrophage phenotypes to an alternatively activated state (**Fig. 5d, e** and **Supplementary Fig. 12**). These data suggest that the effects of beiging on vascular inflammation are biologically relevant.

Comment 5 (minor critique). The authors claim that this effect on vascular remodeling is independent of Ucp1. Succinate metabolism has been considered to mediate thermogenesis independent of Ucp1—might succinate fluxes account for the observed phenotype?

Response 5: We thank the reviewer for this comment. As pointed out by the reviewer, succinate accumulation has been reported to increase thermogenic respiration in brown adipocytes¹⁷. Since PVAT beiging after vascular injury is attenuated by the suppression of β 3AR signaling by the administration of a β 3AR antagonist (SR59230A), we assume that PVAT beiging is mainly mediated by the β 3AR pathway (**Supplementary Fig. 3e**). However, the possibility of the significant role of succinate metabolism in the PVAT beiging process after vascular injury cannot be ruled out, and we thus have mentioned the succinate-mediated pathway in the Discussion section (lines 310–315).

Comment 6. The authors need to validate macrophage switching in vivo using quantifying methods such as FACS. While the in vitro cultures provide preliminary evidence, in vivo identification is required.

Response 6: As described in response #4, we used aortic PVAT with drug-induced vascular injury to examine macrophage polarization using FACS after the β 3AR agonist CL316243 treatment. CL316243 administration shifted PVAT macrophage phenotypes to an alternatively activated state in 10-week-old male C57BL/6 mice that underwent vascular injury (**Fig. 5d, e**

and **Supplementary Fig. 12**).

Comment 7-1. Single cell RNA-sequencing: This reviewer is unclear why the author used pre-existing scRNA-seq datasets to arrive at neuregulin 4 upregulation. First, the authors are interested in PVAT but the scRNA sequencing was performed on subcutaneous inguinal WAT.

Response 7-1: We agree with the reviewer's comment and have attempted scRNA-seq using mouse femoral artery PVAT, as suggested. Given the technical difficulty of single-cell isolation from surrounding vascular tissues, we instead performed single-nucleus RNA sequencing by extracting nuclei from the adipocytes of PVAT, according to previous studies that conducted the sequence using adipose tissues¹⁸. However, the sequencing data did not reach sufficient quality for analysis, mainly because of the small sample volume. Femoral PVAT is a very small tissue (0.03-0.1 mg/mouse) compared to other adipose tissues such as inguinal WAT (100 mg/mouse). PVAT from 10 mice that we collected for one sequence was not enough, and presumably many more mice are needed for good quality analysis. Since use of more than dozens of mice would not be realistic, we performed an alternative analysis using publicly available single-cell data, including multiple integrated pipelines and bulk RNA-seq deconvolution, to make the analysis more convincing (**Supplementary Fig. 6–9** and **Supplementary Table 2**). As results of these validation analyses, the clusters expressing gene signatures of beige cells emerged in other datasets of iWAT treated with CL316243 or cold stimulation, and *Nrg4* was identified as a highly expressed secretory factor in these clusters. These results have been included in the revised manuscript (lines 208-219). Furthermore, to investigate the functional role of NRG4 in the vasculature more clearly, we have also included the results of the *in vivo* and *in vitro* experiments. We confirmed that *Nrg4* siRNA treatment exacerbated pathological intimal thickening 14 days after vascular injury when PVAT was treated with pluronic gel containing siRNA against *Nrg4* (**Figure 4n**). *In vitro* experiments showed that recombinant murine NRG4 protein decreased the Cd86/Mrc1 ratio and the mRNA expression levels of inflammatory cytokines in macrophages classically activated (**Supplementary Fig. 10e, f**). In addition, the knockdown of Erb-B2 receptor tyrosine kinase 4 (ErbB4), a known target of NRG4, abolished alternative activation and anti-inflammatory effects in culture media conditioned by PVAT-derived beige adipocytes (**Supplementary Fig. 10g–i**). Taken together, these results suggest that NRG4 secreted from beige PVAT induces a phenotypic shift of macrophages to an alternatively activated state, leading to accelerated resolution of macrophage inflammation and the attenuation of pathological vascular remodeling after injury. These results have been included in the revised manuscript (lines 237–

249).

Comment 7-2. Second, the authors performed bulk RNA sequencing on PVAT after injury, was neuregulin 4 upregulated in those datasets? The authors then show that neuregulin 4 is upregulated 48 hours after injury. Does it remain elevated throughout the injury like Ucp1 in Fig 1i? Is neuregulin 4 sourced from in vivo derived beige adipocytes? It is known that neuregulin 4 is secreted from thermogenic fat cells (PMID: 25401691); thus, it is not surprising that in vitro derived beige adipocytes secrete neuregulin 4. In vivo neuregulin 4 identification is strongly recommended.

Response 7-2: We appreciate the reviewer's comment. While bulk RNA-seq (n = 2 vs. 2) did not show an apparent upregulation of *Nrg4*, qRT-PCR performed on larger samples under the same conditions showed a significant increase in *Nrg4* gene expression levels at both time points of 24 h and 48 h after injury (**Fig. 4e**). Although the results of the bulk RNA-seq were supportive, the number of samples for the sequence was too small and not convincing. Therefore, the heat map in Figure 1 was moved to **Supplemental Fig. 1a**. *In situ* hybridization analysis revealed that *Nrg4* was upregulated in PVAT surrounding injured femoral arteries at 14 days after injury and was expressed within beige adipocytes of PVAT expressing *Ucp1* (**Supplementary Fig. 10a, b**). NRG4 protein was clearly upregulated in PVAT surrounding dissected aorta in humans, consistent with the observation in murine wire-induced vascular injury model (**Supplementary Fig. 11a [bottom]**). These results suggest that *Nrg4* is upregulated in PVAT throughout the injury and expressed in *Ucp1*-positive thermogenic adipocytes.

Comment 7-3. Third, the scRNA-sequencing was performed under CL316243 conditions; however, the authors are interested in vascular injury induced beige adipogenesis which may induce beige adipocytes differently or have altered beige adipocyte metabolomes etc. These differences need to be reconciled or not shown.

Response 7-3: As described in the response to #3a, PVAT beiging during injury is likely caused by β 3AR activation, supported by the evidence that M2 macrophages accumulating around blood vessels induce PVAT beiging and that the beiging in PVAT elicited by vascular injury is attenuated by inhibition of β 3AR signaling using a β 3AR antagonist (SR59230A) (**Supplementary Fig. 3e**).

Comment 8. The authors demonstrate that in vitro derived PVAT induced beige adipocytes

silenced against neuregulin 4 alters influence macrophage polarity. However, these data are insufficient and incomplete to determine if neuregulin 4 regulates macrophage polarity under these conditions. And these studies were performed using immortalized PVAT preadipocytes, not primary cells. The effects on gene expression are modest at best and again may not be biologically sufficient. Macrophage functional tests should be performed. The authors should inject mice with recombinant Nrg4 protein after vascular injury and assess macrophage polarity and number by flow cytometry and inflammatory and macrophage gene expression. In addition, the authors should use the pluronic gel (as done with Prdm16 Fig 2) with neuregulin 4 siRNA to demonstrate PVAT specificity in vivo. The best outcome would be to generate a genetic mouse model to alter Nrg4 expression to assess if beige adipocytes secrete Nrg4 to regulate vascular remodeling. But I understand this mouse model may be unattainable.

Response 8: As pointed out by the reviewer, the analysis evaluating the biological function of *Nrg4* in the vascular response to injury is critical. To address this matter, we have included several experiments as follows: In an *in vitro* study, recombinant murine NRG4 decreased the *Cd86/Mrc1* ratio and the expression levels of inflammatory cytokines in classically activated macrophages (**Supplementary Fig. 10e, f**). In addition, the knockdown of Erb-B2 receptor tyrosine kinase 4 (ErbB4), a receptor of NRG4, in macrophages abolished alternative activation and anti-inflammatory effects of culture media conditioned by PVAT-derived beige adipocytes (**Supplementary Fig. 10g–i**), suggesting the importance of *Nrg4*-ErbB4 signaling in demonstrating the anti-inflammatory effects of *Nrg4*. Furthermore, consistently with RAW 264.7 cells, culture media conditioned by PVAT-derived beige adipocytes induced phenotypic changes from classical to alternative activation and downregulation of inflammatory genes in bone marrow-derived monocytes (BMDM) (**Fig. 3c, d**). These changes were not observed in culture media conditioned by PVAT-derived beige adipocytes with the knockdown of *Nrg4* by siRNA (**Fig. 4k, l**). Finally, the knockdown of *Nrg4* in perivascular tissue exacerbated vascular remodeling after injury, as shown in **Figure 4n**. Taken together, these results suggest that *Nrg4* upregulation in beige PVAT after injury is functionally relevant in the regulation of the inflammatory response and subsequent vascular remodeling.

Comment 9. The AAD modeling in humans is lacking information to make formal conclusions based on the observed phenotype. First, are the samples from male or female adult humans? Second, what were the inclusion and exclusion criteria for the AAD patients (sex, age, disease states). AAD can be caused by several disease states or syndromes. It is not clear if the patients had hypertension, atherosclerosis, or other cardiovascular diseases, predisposing them to AAD.

Third, were the biopsies collected at the time of AAD or follow-up? If these sample were collected at the time of AAD it would appear these data are contradictory to the authors hypothesis. The patients would be suffering from other cardiometabolic diseases predisposing them to vascular inflammation and the impinging aortic tear. Thus, based upon the data presented by the authors there should be less beige fat or functionally disrupted beige adipocytes, as its ability to protect the patients from AAD would be reduced? Therefore, based on the data concluded from the authors I would have expected less or dysfunctional beige adipose tissue in AAD patients.

Response 9: As suggested by the reviewer, we mentioned information about humans of whom aorta we examined. The aortic specimens of AAD patients were autopsy specimens from six patients who died of AAD: five males and one female, aged between 45 and 84 years. Four patients had hypertension, one patient had a history of stroke, and three patients had a history of smoking. The aortic specimens of the controls were autopsy specimens from five patients who died of non-aortic causes, four males and one female, aged between 44 and 78 years. One patient had hypertension, one patient had dyslipidemia, one patient had chronic kidney disease, one patient had a history of stroke, and one patient had a history of smoking. The inclusion and exclusion criteria for AAD patients were as follows: AAD cases were selected from consecutive pathological autopsies from 2006 to 2017 with a description of aortic dissection as a finding in autopsy report. Controls were selected from consecutive pathological autopsies from 2014 to 2017 with no significant changes in the aortic wall based on the autopsy. Patients with an apparent history of inherited aortic disease, such as Marfan syndrome, were not included in the AAD cases and controls. This description has been included in the revised manuscript (lines 519–531).

As the reviewer pointed out, it is not possible to determine whether PVAT beiging is a cause or consequence of vascular injury from the analysis of human pathological specimens alone. Thus, in addition to wire-induced vascular injury model, we used a murine AAD model as a model pathophysiologically relevant to humans. The activation of beiging by CL316243 administration significantly reduced death due to dissection or rupture of the aorta (**Fig. 5c**), concomitant with the shift of macrophage phenotypes in PVAT to the alternatively activated state evaluated by fluorescence-activated cell sorting (FACS) analysis (**Fig. 5d, e** and **Supplementary Fig. 12**). Based on these findings in humans and mice, we considered that a certain level of PVAT beiging is an intrinsic stress response to vascular injury, contributing to the attenuation of excessive vascular inflammation.

Comment 10. These data hint at an interesting physiology under vascular remodeling and injury; however, it is unclear if the failure to induce beige adipocytes results in poor prognosis or development of vascular diseases. The authors need to identify a pathophysiological model to determine if beige adipose tissue influences vascular disease. Such suggested models include diet-induced obesity or ApoE-null mice.

Response 10: According to the reviewer's suggestion, we performed experiments using an acute aortic dissection (AAD) murine model, which was induced by the simultaneous administration of angiotensin II, β -aminopropionitrile, and N-nitro-L-arginine methyl ester to 10-week-old male C57BL/6 mice. CL316243 administration shifted PVAT macrophage phenotypes into an alternatively activated state, which was confirmed by fluorescence-activated cell sorting (FACS) analysis (**Fig. 5d, e** and **Supplementary Fig. 12**), and significantly reduced deaths due to dissection or rupture of the aorta (**Fig. 5c**), suggesting that the induction of beiging is physiologically relevant and improves the prognosis of acute vascular diseases such as AAD.

Comment 11. The authors state that Ucp1 is involved in atherogenesis (ref 30, 32-35). However, their data suggest that Ucp1 does not mediate vascular remodeling after injury but beige fat—characterized by Ucp1 expression and function—is critical for injury remodeling. Again, the authors state that inflammation plays a critical role in the development of atherosclerosis and is one of the major reasons to initiate these studies. The authors should further discuss this topic to speculate how their data rederive differences between their study and published studies. Moreover, is neuregulin 4 secreted from beige adipocytes void of Ucp1?

Response 11: As pointed out by the reviewer, the role of *Ucp1* in atherogenesis has been studied previously. The *Ucp1* deletion in ApoE-KO mice showed attenuated atherosclerotic plaque burden¹⁹(ref 32 in the initial manuscript), while the *Ucp1* overexpression specifically in vascular smooth muscle cells exacerbates atherosclerosis²⁰(ref 34 in the initial manuscript). In our study, the *Ucp1* deletion showed a trend toward attenuation of neointimal hyperplasia (pathological vascular remodeling), although the difference was not statistically significant (**Fig. 3h**). These findings suggest that the effects of *Ucp1* itself on the vasculature may be detrimental at least in the chronic progression of atherosclerosis and possibly in the acute vascular injury response. Meanwhile, beiging stimulation with CL316243 reportedly improves atherogenesis, although the expression levels of *Ucp1* in adipose tissues were concomitantly upregulated by the treatment²¹ (ref 30 in the initial manuscript). We thus believe that factors other than *Ucp1* induced by PVAT beiging may overcome the potential negative effects of *Ucp1*

and improve vascular remodeling, and *Nrg4* identified in this study may be one of the potent vasoprotective factors induced by PVAT beiging.

As the reviewer suggested, we discussed the role of inflammation in the development of atherosclerosis, focusing on the differences between published studies and ours. Various previous studies have suggested that endothelial dysfunction leads to the recruitment of circulating monocytes to the intima–medial layer^{22,23}. We observed that the marked accumulation of monocytes was also identified in the outer tissues surrounding the vasculature from the early phase after vascular injury (**Fig. 1b, c**). The infiltrated monocyte-macrophages elicited PVAT beiging (**Fig 1k, l**), resulting in the suppression of the excessive vascular inflammatory response to the injury (**Fig 2a-d**). These data indicate a pivotal role of the “outside-in” manner in the wound healing process after vascular injury. We have addressed this point in the Discussion section of the revised manuscript (lines 279–285).

Finally, *Nrg4* expression after vascular injury in the PVAT of *UCP1*^{-/-} mice was comparable to that of *UCP1*^{+/+} mice (**Supplementary Fig. 10c**), indicating that *Nrg4* is expressed in a *Ucp1*-independent manner.

Reviewer #3 (Remarks to the Author):

This new study from Adachi et al. examine the role of perivascular adipose tissue (PVAT) being in the regulation of vascular inflammation and remodeling. The present study utilizes a genetic mouse model in which adipose tissue cannot undergo “browning” (adipocyte Prdm16 knockout) in the setting of endovascular injury-induced inflammation. The data support a hypothesis that the macrophage-induced being of PVAT is needed to amount an eventual counterregulatory inflammatory response. The authors propose that the beige adipocyte derived NRG4 mediates this anti-inflammatory response by promoting the alternative anti-inflammatory phenotype of macrophages.

Overall, the paper is well written and the data are robust and important for the field. The paper nicely highlights a role for beige adipocytes in the resolution of injury-induced inflammation, involving a mechanism distinct from its classical thermogenic (UCP1-mediated) function. Despite these strengths, there are some weaknesses to current study that require attention.

Comment 1. Evidence that Nrg4 is a key regulatory molecule that mediates the effect of being in vivo is lacking. The authors are encouraged to knockout Nrg4 in vivo in a manner analogous to what was done for Prdm16 (e.g. siRNA). Also, the study of human AAD samples would be stronger if Nrg4 expression could be shown (e.g. histology).

Response 1: As suggested by the reviewer, to examine the functional role of *Nrg4* in the vasculature, we have added the following experiments. We reduced the expression levels of *Nrg4* in PVAT by the treatment with *siNrg4*. Vascular remodeling after injury was significantly exacerbated in the *siNrg4*-treated group compared to the control group (**Fig. 4n**). In addition, we examined the expression of NRG4 in human aorta with or without dissection. The expression of NRG4 was significantly upregulated in human AAD PVAT (**Supplementary Fig.11a** [bottom]).

Comment 2. A recent study by Seale et al. (Nature Metabolism 2021) provided a nice scRNA analysis of PVAT. The authors should discuss their findings with respect to this new literature. Some comparison of the datasets would be helpful.

Response 2: Thank you for the reviewer’s insightful comments. Seale *et al.* performed an excellent scRNA analysis of human and murine PVAT¹³ (*Nature Metabolism* 2021) and revealed that PVAT beige adipocytes are differentiated from a smooth muscle-like origin. They also observed in the study an increase in MYH11⁺ smooth muscle cell-derived adipocytes in PVAT after treatment with rosiglitazone, known to recruit beige cells in fat tissues¹⁴.

Consistently, another study clearly showed that in SM22-expressing cell-specific peroxisome proliferator-activated receptor- γ knockout mice, the disruption of adipogenesis in smooth muscle cells causes hypoplasia of PVAT, nevertheless other adipose tissues such as WAT and BAT were intact¹². Based on these observations, we developed *Tagln* (Transgelin, coding SM22)-*Cre*^{+/-};*Prdm16*^{fllox/fllox} mice, referred to as *Tagln*^{Cre+};*Prdm16* mice, in which beiging is inhibited in SM22-positive smooth muscle cell lineage. PVAT beiging was significantly attenuated in PVAT surrounding thoracic aorta in *Tagln*^{Cre+};*Prdm16*^{fllox/fllox} mice compared with *Tagln*^{Cre-};*Prdm16*^{fllox/fllox} control mice (**Figure 1. for reviewer #3**). This result suggests that beiging cells in PVAT belong to a lineage of SM22-positive smooth muscle cells. We referred to the literature provided by the reviewer and discussed in the Discussion session (lines 303–310).

Figure 1. for reviewer #3: Gene expression levels of BAT (*Ucp1* and *Prdm16*) and WAT [*Cfd* (Adipsin) and *Rstn* (Resistin)] markers in PVAT surrounding thoracic aorta in *Tagln*^{Cre+};*Prdm16* mice compared with *Tagln*^{Cre-};*Prdm16* control littermates (n=3–5 for each group, multiple *t*-tests with Holm-Sidak's correction). Data represent mean \pm SEM **P* < 0.05, ***P* < 0.01.

Reviewer #4 (Remarks to the Author):

Adachi and Ueda et al reported that beiging of perivascular adipose tissue protected vascular injury through modulating inflammatory responses and vascular remodeling. To support this conclusion, the authors performed a number of analyses on the data collected from web lab-based experiments and public genomics data repository. Since my expertise is in bulk and single cell transcriptomics but not in adipogenesis, my comments will be largely restricted to this area. Overall, the bulk RNA-seq and single cell RNA-seq data present in this manuscript are not strong enough to support the main conclusion. Here are my comments for consideration.

Comment 1. The bulk RNA-seq data present in Figure 1e only contains n=2 from each group (sham versus injury). No statistics can be done on this low sample size so the data from this figure is not convincing. The authors are strongly advised to either increase the N or remove the figure.

Response 1: Along with the reviewer's comment, the bulk RNA-seq data have been removed from the main figure and moved to **Supplementary Fig. 1a**.

Comment 2. The single nucleus RNA-seq data shown in Figure 4 and Extended Figures 3-4 were not generated by this study but from re-analysis of a public dataset (GSE133486). Several issues might arise from re-analysis of the data from other studies. First, the sample types are different. In the original paper, the snRNA-seq data were obtained from inguinal white adipose tissue (iWAT) and not PVAT. It is unclear if the adipocyte heterogeneity is different from tissue to tissue. Second, the authors only selected n=1 sample from each group to complete the analysis. This low sample size analysis, again, will compromise any conclusions drawn from it. Third, the authors identified a cluster from the analysis that only comes from the injury group. That would also happen when the integration algorithm cannot robustly correct for batch effect. The authors should apply multiple integrated pipelines (Seurat, Liger, Harmony, MNN, etc) on the same data and see if the cluster is from technical artifact of the integration or is a real cell type. I strongly suggest that the authors should create their own single cell data from PVAT to demonstrate the transition of the adipocytes from normal to beiging state during vascular injury. However, if the single cell data is not critical in this study or other experimental data are sufficient to support the main conclusions, I have the following suggestions which might help strengthen the current analysis.

Response 2: We have attempted single-nucleus RNA-seq using mouse femoral artery PVAT, as recommended by the reviewer. However, the sequencing data did not reach sufficient quality

for analysis, mainly because of the small sample volume. Femoral PVAT is an exceedingly small tissue (0.03-0.1 mg/mouse) compared to other adipose tissues such as inguinal WAT (100 mg/mouse). PVAT from 10 mice that we collected for one sequence was not enough, and presumably many more mice are needed for good quality analysis. Thus, we performed an alternative analysis using publicly available single-cell data to make our findings more convincing, as the reviewer suggested. **Supplementary Fig. 6a–c** shows the clustering analysis of UMAP dimensionality reduction using scRNA-seq data in publicly available datasets of mouse inguinal WAT treated with β 3-AR agonist CL316243 or control (GSE 133486)². The integration step was performed using Harmony (V.0.1)²⁴. Adipose tissue cells were divided into 16 clusters that were color-coded by cell type (**Supplementary Fig. 6b**) and stimulation (CL316243 treatment [pink] or control [turquoise]) (**Supplementary Fig. 6a**). As indicated in **Supplementary Fig. 6b, c**, cluster 6 was predominantly composed of CL316243-treated cells. The violin plots in **Supplementary Fig. 6d** showed accumulation of representative beige/BAT marker genes in C6. These results show that a characteristic beige cluster appeared after CL316243 administration even with the integration step using Harmony, supporting that this cluster is not a technical artifact of the integration but rather a real cell type.

As pointed out by the reviewer, *Nrg4* we identified in the snRNA-seq data was obtained from inguinal WAT but not from PVAT. Thus, we carefully assessed the functional role of *Nrg4* in vascular remodeling and added the results of *in vivo* and *in vitro* experiments. We confirmed that *Nrg4* siRNA treatment exacerbated pathological intimal thickening 14 days after vascular injury (**Fig. 4n**). Recombinant murine NRG4 protein reduced the Cd86/Mrc1 ratio and the mRNA expression levels of inflammatory cytokines in classically activated macrophages *in vitro* (**Supplementary Fig. 10e, f**). In addition, the knockdown of Erb-B2 receptor tyrosine kinase 4 (ErbB4), a receptor of NRG4, abolished alternative activation and anti-inflammatory effects of culture media conditioned by PVAT-derived beige adipocytes (**Supplementary Fig. 10g–i**). Taken together, these results suggest that NRG4 secreted from beige PVAT induces a phenotypic shift of macrophages to an alternatively activated state, leading to accelerated resolution of macrophage inflammation and attenuation of pathological vascular remodeling after injury. We have included these results in the revised manuscript (lines 208–211, 237–245).

Comment 3. A number of bulk RNA-seq data (e.g. PMID:32139607 and PMID: 30116775) were published to study the beiging of white adipose tissue. The authors can use the gene

signature from the single cell data to deconvolute these public bulk RNA-seq data to confirm that the beige adipocyte also exists in the injured WAT from other studies. Computational algorithms for cell type deconvolution include MuSiC, BSEQ-sc, Bisque, etc. This cross validation approach would make the data more convincing.

Response 3: We agree with the reviewer's comment. The gene signature from the public scRNA-seq data (GSE133486)² was used to deconvolute the public bulk RNA-seq data with the MuSiC²⁵ to confirm the validity of the cell type clustering performed in the scRNA-seq public data re-analysis. Beige adipocytes also emerged in inguinal WAT by CL316243 (GSE129083²⁶; **Supplementary Table 2** and GSE86338²⁷; Table 1. for Reviewer #4) or cold stimulation (GSE72603²⁸; Table 2. for Reviewer #4). We have included these results in the revised manuscript (lines 211–213).

Comment 4. Two single cell RNA-seq datasets on beige WAT from mouse (PMID: 29937373) and human (PMID: 32066997) are also publicly available. The authors can re-analyze these two datasets and see whether the same cell states are present in these two studies.

Response 4: Thank you for the reviewer's comment. We re-analyzed single-cell RNA-seq datasets²⁹ (PMID: 29937373) and found that *Ucp1*-positive beige cells were seldom observed even after CL316243 stimulation. This is likely because these data were obtained from mouse adipose tissue stromal cells and not mature adipocytes. Stromal cells have distinct characteristics from mature adipocytes, which was described as the study limitation of that paper²⁹. Therefore, it was difficult to compare these datasets in the present study. In fact, although the scRNA-seq datasets² used in our study also contain single-cell data from the murine stromal vascular fraction, the expression of beige fat markers was seldom observed. Another single-cell RNA-seq dataset³⁰ (PMID: 32066997) was also used to analyze the human stromal vascular fraction. However, the data did not include stimulated conditions, such as CL316243 or cold exposure. At present, the public scRNA-seq dataset² used in this study is the only available single-cell RNA-seq data from mature adipose tissue compared between the control and stimulated conditions (CL316243 or cold stimulation).

*Comment 5. The author can perform ligand-receptor analysis (using CellPhoneDB, etc) on the single cell RNA-seq data to delineate the interactions between beige adipocyte and immune cells or endothelial cells are mediated through *Nrg4* and its associated receptors.*

Response 5: We appreciate the reviewer's valuable recommendations. Ligand-receptor analysis using CellChat³¹ on single-cell RNA-seq data (GSE133486)² identified endothelial

cells as a cluster interacting with beige adipocytes (**Figure 1. for Reviewer #4**), but this was considered challenging to explain the molecular mechanism associated with NRG4. A possible reason was that the expression level of *ErbB4*, a receptor for NRG4, was overall low in the single-cell dataset. In our experimental studies, the administration of recombinant murine NRG4 changed macrophage polarity to an anti-inflammatory type (**Supplementary Fig. 10e, f**), and *Nrg4* knockdown shifted macrophage polarity to an inflammatory type (**Fig. 4i-l, Supplementary Fig. 10d**). *ErbB4* knockdown shifted macrophages to an inflammatory subtype and enhanced the expression levels of inflammatory cytokines (**Supplementary Fig. 10g-i**). These results suggest that the NRG4-ErbB4-mediated pathway contributes to improving inflammation, which is consistent with the previous study showing that macrophages are directed towards anti-inflammation via the NRG4-ErbB4 pathway^{6,7}.

Figure 1. for reviewer #4: A circle plot visualizing the communication network of each cluster of single-cell RNA-seq data is shown. Circle size and edge width are proportional to the number of cells in each cell cluster and the communication score between interacting cell clusters, respectively.

References

1. Mu, W., *et al.* BMP4-mediated browning of perivascular adipose tissue governs an anti-inflammatory program and prevents atherosclerosis. *Redox Biol* **43**, 101979 (2021).
2. Rajbhandari, P., *et al.* Single cell analysis reveals immune cell-adipocyte crosstalk regulating the transcription of thermogenic adipocytes. *Elife* **8**(2019).
3. Geissler, A., Ryzhov, S. & Sawyer, D.B. Neuregulins: protective and reparative growth factors in multiple forms of cardiovascular disease. *Clin Sci (Lond)* **134**, 2623-2643 (2020).
4. Wang, G.X., *et al.* The brown fat-enriched secreted factor Nrg4 preserves metabolic homeostasis through attenuation of hepatic lipogenesis. *Nat Med* **20**, 1436-1443 (2014).
5. Guo, L., *et al.* Hepatic neuregulin 4 signaling defines an endocrine checkpoint for steatosis-to-NASH progression. *J Clin Invest* **127**, 4449-4461 (2017).
6. Schumacher, M.A., *et al.* NRG4-ErbB4 signaling represses proinflammatory macrophage activity. *Am J Physiol Gastrointest Liver Physiol* **320**, G990-G1001 (2021).
7. Schumacher, M.A., *et al.* ErbB4 signaling stimulates pro-inflammatory macrophage apoptosis and limits colonic inflammation. *Cell Death Dis* **8**, e2622 (2017).
8. Long, J.Z., *et al.* A smooth muscle-like origin for beige adipocytes. *Cell Metab* **19**, 810-820 (2014).
9. Berry, D.C., Jiang, Y. & Graff, J.M. Mouse strains to study cold-inducible beige progenitors and beige adipocyte formation and function. *Nat Commun* **7**, 10184 (2016).
10. Rosenwald, M., Perdikari, A., Rulicke, T. & Wolfrum, C. Bi-directional interconversion of brite and white adipocytes. *Nat Cell Biol* **15**, 659-667 (2013).
11. Roh, H.C., *et al.* Warming Induces Significant Reprogramming of Beige, but Not Brown, Adipocyte Cellular Identity. *Cell Metab* **27**, 1121-1137 e1125 (2018).
12. Chang, L., *et al.* Loss of perivascular adipose tissue on peroxisome proliferator-activated receptor-gamma deletion in smooth muscle cells impairs intravascular thermoregulation and enhances atherosclerosis. *Circulation* **126**, 1067-1078 (2012).
13. Angueira, A.R., *et al.* Defining the lineage of thermogenic perivascular adipose tissue. *Nat Metab* **3**, 469-484 (2021).
14. Ohno, H., Shinoda, K., Spiegelman, B.M. & Kajimura, S. PPARgamma agonists induce a white-to-brown fat conversion through stabilization of PRDM16 protein. *Cell Metab* **15**, 395-404 (2012).
15. Nguyen, K.D., *et al.* Alternatively activated macrophages produce catecholamines to sustain adaptive thermogenesis. *Nature* **480**, 104-108 (2011).
16. Wang, Y.N., *et al.* Slit3 secreted from M2-like macrophages increases sympathetic activity and thermogenesis in adipose tissue. *Nat Metab* **3**, 1536-1551 (2021).
17. Mills, E.L., *et al.* Accumulation of succinate controls activation of adipose tissue thermogenesis. *Nature* **560**, 102-106 (2018).
18. Van Hauwaert, E.L., *et al.* Isolation of nuclei from mouse white adipose tissues for single-nucleus

- genomics. *STAR Protoc* **2**, 100612 (2021).
19. Dong, M., *et al.* Cold exposure promotes atherosclerotic plaque growth and instability via UCP1-dependent lipolysis. *Cell Metab* **18**, 118-129 (2013).
 20. Bernal-Mizrachi, C., *et al.* Vascular respiratory uncoupling increases blood pressure and atherosclerosis. *Nature* **435**, 502-506 (2005).
 21. Berbée, J.F., *et al.* Brown fat activation reduces hypercholesterolaemia and protects from atherosclerosis development. *Nat Commun* **6**, 6356 (2015).
 22. Ross, R. & Glomset, J.A. The Pathogenesis of Atherosclerosis. *New England Journal of Medicine* **295**, 369-377 (1976).
 23. Rader, D.J. & Daugherty, A. Translating molecular discoveries into new therapies for atherosclerosis. *Nature* **451**, 904-913 (2008).
 24. Korsunsky, I., *et al.* Fast, sensitive and accurate integration of single-cell data with Harmony. *Nat Methods* **16**, 1289-1296 (2019).
 25. Wang, X., Park, J., Susztak, K., Zhang, N.R. & Li, M. Bulk tissue cell type deconvolution with multi-subject single-cell expression reference. *Nat Commun* **10**, 380 (2019).
 26. Wang, W., *et al.* A PRDM16-Driven Metabolic Signal from Adipocytes Regulates Precursor Cell Fate. *Cell Metab* **30**, 174-189 e175 (2019).
 27. Bai, Z., *et al.* Dynamic transcriptome changes during adipose tissue energy expenditure reveal critical roles for long noncoding RNA regulators. *PLoS Biol* **15**, e2002176 (2017).
 28. Xue, H., *et al.* Molecular signatures and functional analysis of beige adipocytes induced from in vivo intra-abdominal adipocytes. *Sci Adv* **4**, eaar5319 (2018).
 29. Burl, R.B., *et al.* Deconstructing Adipogenesis Induced by beta3-Adrenergic Receptor Activation with Single-Cell Expression Profiling. *Cell Metab* **28**, 300-309 e304 (2018).
 30. Vijay, J., *et al.* Single-cell analysis of human adipose tissue identifies depot and disease specific cell types. *Nat Metab* **2**, 97-109 (2020).
 31. Jin, S., *et al.* Inference and analysis of cell-cell communication using CellChat. *Nat Commun* **12**, 1088 (2021).

Reviewers' Comments:

Reviewer #1:

Remarks to the Author:

I am happy with the changes that have been made.

Reviewer #2:

Remarks to the Author:

Thank you for addressing my comments. Everything has been addressed and the conclusion within manuscript have been strengthened from the previous version.

Reviewer #3:

Remarks to the Author:

The authors have provided new data that address this reviewer's concerns. This paper should be a nice addition to the field.

Reviewer #4:

Remarks to the Author:

The authors have addressed the concerns I raised for the single cell data analysis. The manuscript has been improved in the revision.